# Neonatal brain injury causes cerebellar learning deficits and Purkinje cell dysfunction

Aaron Sathyanesan [1], Srikanya Kundu [1], Joseph Abbah[1] & Vittorio Gallo[1,2]

Premature infants are more likely to develop locomotor disorders than term infants. In a chronic sub-lethal hypoxia (Hx) mouse model of neonatal brain injury, we recently demonstrated the presence of cellular and physiological changes in the cerebellar white matter. We also observed Hx-induced delay in Purkinje cell (PC) arborization. However, the behavioral consequences of these cellular alterations remain unexplored. Using the Erasmus Ladder to study cerebellar behavior, we report the presence of locomotor malperformance and long-term cerebellar learning deficits in Hx mice. Optogenetics experiments in Hx mice reveal a profound reduction in spontaneous and photoevoked PC firing frequency. Finally, treatment with a gamma-aminobutyric acid (GABA) reuptake inhibitor partially rescues locomotor performance and improves PC firing. Our results demonstrate a long-term miscoordination phenotype characterized by locomotor malperformance and cerebellar learning deficits in a mouse model of neonatal brain injury. Our findings also implicate the developing GABA network as a potential therapeutic target for prematurity-related locomotor deficits.

[1] Center for Neuroscience Research, Children's Research Institute, Children's National Medical Center, Washington, DC, USA. [2] The George Washington University School of Medicine and Health Sciences, Washington, DC, USA. These authors contributed equally: Aaron Sathyanesan, Srikanya Kundu. Correspondence and requests for materials should be addressed to V.G. (email: vgallo@childrensnational.org)

Premature birth impedes brain growth and development that normally occurs in the third trimester of gestation. The cerebellum especially sees rapid development during late gestation, almost quadrupling in size[1], with the surface of the cerebellar cortex increasing 30 times with concurrent increased foliation[2,3]. At the cellular level, the Purkinje cell (PC) layer as well as the external granule cell layer exhibit continued differentiation and rapid cell division, respectively[4]. Thus, premature birth drastically affects cerebellar development and associated behavior[5]. This is borne out by studies in preterm human subjects documenting the presence of severe motor and cognitive disabilities, spanning infancy[6] through adolescence[7], to young adulthood[8], and even adulthood[9].

Two increasingly common types of injuries found in the brains of premature infants are diffuse white matter injury and decreased volume of the cerebral cortex (gray matter). An established cause of both grey and white matter abnormalities in the premature brain is hypoxic injury owing to a vulnerable and underdeveloped respiratory system. As the cerebellum is rapidly developing during late gestation, it is particularly vulnerable to hypoxic injury[3]. That the cerebellar cortex suffers hypoxic insult in premature infants is bolstered by several animal model studies[10–14]. In rodents, the first two postnatal weeks of cerebellar development translate to the human cerebellar developmental timeline during the third trimester. During this developmental time window, multiple, transient, circuit-level modifications occur. Significant developmental modifications involve PCs, which are the principal neurons of the cerebellar cortex[15]. It is thus generally understood that proper PC circuit maturation is required for normal cerebellar behavior at later stages[16]. However, the relationship between alterations in particular aspects of PC physiology and abnormalities in cerebellar behavior induced by neonatal brain injury during circuit maturation is not well defined.

We and others, have characterized and utilized a mouse model of chronic perinatal hypoxia (Hx) that faithfully recapitulates the cellular and morphological hallmarks of neonatal brain injury in human infants[17,18]. In this model, mice are reared in a low oxygen environment from postnatal day 3 through 11 (P3–P11), resulting in gray and white matter volume reduction, typical of neurodevelopmental delay[19]. This is followed by a period of recovery, wherein cortical volumes reach normoxic (Nx) levels by adulthood. However, some aspects of hypoxic brain injury persist, including dysmaturation of specific neuronal and glial cell populations in both gray and white matter regions of the brain, and disruption of inhibitory gamma-aminobutyric acid (GABA) neurotransmission[20] similar to what is observed in humans[21]. Evidence obtained by multiple laboratories strongly indicates that cerebellar abnormalities result at least in part from defective GABA signaling involving both neurons and glia[22,23] in the adult brain. In the developing brain, GABA plays distinct roles including neuronal excitation[24,25] and trophic signaling[26–28]. However, potential disruption of neurodevelopmental GABA signaling as a causative factor of behavioral abnormalities owing to neonatal brain injury has not been investigated.

Recently, we have demonstrated that drastic cellular and physiological changes occur in the cerebellar white matter (WM) following chronic perinatal Hx. Specific alterations include hypomyelination, and changes in physiological profiles of oligodendrocyte precursor cells, as well as GABAergic interneurons[29]. Importantly, we also observed changes in cerebellar gray matter, including a significant loss of GABAergic interneurons and a dramatic post-Hx decrease in dendritic arborization of PCs, whose axons represent the only output of the cerebellar cortex, critical for motor performance and motor learning. These cellular and morphological changes in the cerebellar cortex are associated

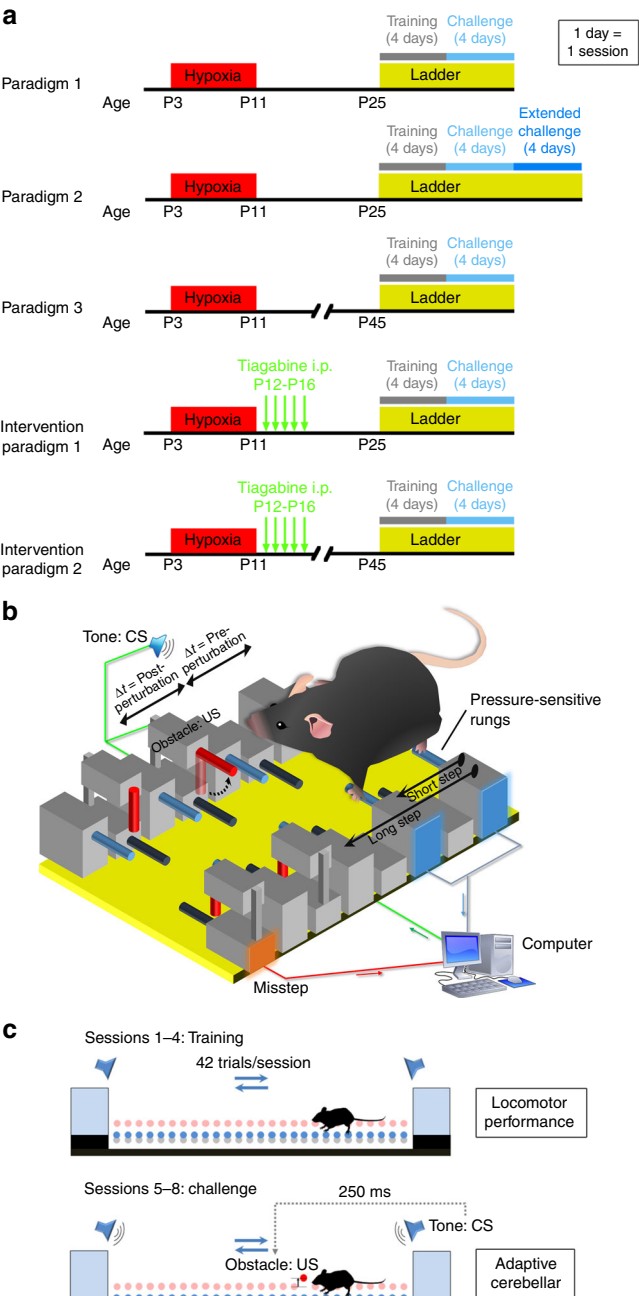

with changes in firing properties of PCs indicating a potential feature of Hx pathophysiology[29].

In the present study, we sought to determine the specific behavioral and in vivo physiological consequences of cerebellar injury due to Hx. Optimal cerebellar function is critical to proper motor performance, as well as efficient motor learning. Therefore, disruptions in cerebellar circuitry lead to motor malperformance and deficits in associative locomotor learning. To identify potential changes in cerebellar behavior, we used the Erasmus ladder (Fig. 1), which is a fully automated and computerized behavioral apparatus that allows detailed analysis and quantification of motor performance and cerebellar locomotor learning in mice[30] under normal physiological conditions or after injury, e.g., after chronic perinatal Hx. Our results indicate that—2 weeks after Hx—mice perform significantly worse than Nx controls in both motor performance, as well as conditioned, adaptive motor learning. Delayed learning is still detectable at 5 weeks after Hx.

**Fig. 1** Experimental design and schema to test cerebellar-mediated behavior in mice with neonatal brain injury. **a** Paradigm to test for Hx-induced changes in P25 mice using standard protocol (Paradigm 1), and extended challenge protocol (Paradigm 2). P45 mice were tested using Paradigm 3. Intervention paradigms 1 and 2 were used to test pharmacological intervention for P25 Hx and P45 Hx mice respectively. Hx timeline (P3–P11) (red box, "Hypoxia"), Erasmus ladder experiment (yellow box, "Ladder") divided into training (session 1–4, grey bar), followed by challenge sessions to test adaptive cerebellar learning (sessions 5–8, light blue bar). For paradigm 2, additional four sessions of challenge were included (sessions 9–12, deep blue bar). Green arrows in Intervention paradigms 1 and 2 represent individual injections of Tiagabine i.p. from P12–P16 (one injection/day). **b** Cartoon (not drawn to scale) depicting a small portion of the Erasmus Ladder during mouse locomotion. Pressure-sensitive default walking rungs and misstep rungs are shown in dull blue and dark blue, respectively. Stepping patterns (short/long) are captured as mouse traverses the ladder. During challenge sessions to test adaptive learning, a high-pitched tone (CS) is played, followed by random obstacle presentation (US). Steptime changes for rungs preceding the obstacle (Pre-perturbation) and between immediately-preceding rung and immediately-after rung (Post-perturbation) are used to determine learning. **c** Side view schematic of Erasmus Ladder depicting sessions 1–4 primarily measuring locomotor performance and sessions 5–8 primarily measuring adaptive cerebellar learning. Mouse (black), default walking rungs (blue), active obstacle (red), inactive obstacles (salmon), goal boxes (light blue)

As abnormal cerebellar learning and locomotor behavior is directly related to developmental and physiological integrity of PCs, we conducted in vivo optogenetics experiments coupled with multielectrode array recordings in Hx animals. Our electrophysiological experiments reveal profound changes in both spontaneous and photoevoked PC firing. Finally, pharmacological treatment with Tiagabine—a GABAergic reuptake inhibitor that elevates GABA levels in brain—partially ameliorates locomotor performance and improves PC firing frequency. Our study systematically quantifies and treats specific cerebellar behavioral deficits owing to Hx, and provides a potential link to pathophysiology in an animal model of neonatal brain injury.

## Results

**Neonatal brain injury causes locomotor miscoordination.** As cerebellar injury in humans results in locomotor miscoordination[31,32], we tested for locomotor coordination deficits in mice at 2 weeks following hypoxic rearing (P25). On training sessions 1 and 2, Hx mice displayed severe motor malperformance compared with Nx mice (Fig. 2a). On average, Nx control mice committed only $1.7 \pm 0.3\%$ (SEM, $n = 8$ animals) missteps in session 1, whereas Hx mice committed an average of 17% missteps $\pm 2.1\%$ ($n = 11$ animals). In session 2, whereas both Hx and Nx mice committed lesser missteps compared with their previous performance in session 1, Hx mice still committed significantly higher number of missteps ($6.7 \pm 1.4\%$ SEM) compared with Nx controls for which misstep rate dropped to $< 1\%$ ($0.7 \pm 0.2\%$ SEM) ($P < 0.0001$). However, this difference in percentage of missteps between Hx and Nx was statistically insignificant from session 3 till the end of training ($P > 0.05$ in all comparisons). Even in challenge sessions 5 through 8, when Hx mice faced an obstacle paired with a preceding warning tone, their misstep percentage were relatively less compared with session 1 (Fig. 2a). Contrasted to the difference in percentage of backsteps, which can be viewed as a measure of motivation to complete the trial, differences in misstep percentage was larger between the two groups (Fig. 2a, b). Multiple comparison tests reveal that the only session where a statistically significant difference was found

between Hx and Nx backstep percentage was in session 2, with Hx mice registering less than an average of 1.5% backsteps (Fig. 2b). This shows that differences in motor performance, are appropriately indicated by higher percentage of missteps in Hx, as differences in motivation between Hx and Nx are minimal. Thus, although Hx mice start off with higher baseline missteps, they eventually perform relatively well compared with Nx controls.

**Neonatal brain injury causes cerebellar learning deficits.** Cerebellar damage specifically alters the ability to learn adaptive locomotor movements in human patients[33]. Hence, we tested for adaptive cerebellar learning deficits in Hx mice using the Erasmus Ladder system. During sessions 1 through 4 (training), post-perturbation steptimes of Hx and Nx groups were not significantly different (Fig. 3a). However, Hx mice which began training at P25 displayed a striking deficit in cerebellar learning, as evidenced by significant differences between the post-perturbation steptimes (solid red trace) compared with Nx controls (solid black trace) in session 5 (Fig. 3a). This difference in post-perturbation steptime persisted in subsequent challenge sessions. At the end of the learning paradigm, younger Hx mice still had significantly higher post-perturbation steptimes compared with age-matched Nx mice (Fig. 3a).

As these measurements are absolute comparisons, it is possible that although these may be significantly different, relative changes in learning may still be insignificant. To compare relative changes in learning, we normalized post-perturbation steptimes for an individual mouse to post-perturbation steptimes recorded for the same mouse on the last day of training (Fig. 3b). Although a normalized learning value equal to or less than zero indicates higher relative learning, a value higher than zero indicates lower relative learning. Using these metrics, we observed that mean normalized learning in the Hx group consistently stayed above zero, indicating lower relative learning, whereas Nx animals have significantly lower normalized values indicating better relative learning (Fig. 3b).

In addition, to comparing steptimes across groups, we also compared pre-perturbation and post-perturbation steptimes within groups. We discovered that within both Nx (black traces) and Hx (red traces) groups there are statistically significant differences between mean pre-perturbation (broken traces) and post-perturbation steptimes (solid traces) during the first challenge session (Fig. 3a). During subsequent sessions, the Nx group showed reduced difference between mean pre-perturbation and post-perturbation steptimes, which remains statistically insignificant from sessions 6 through 8. However, for the Hx group, mean pre-perturbation and post-perturbation statistically significant differences persist till session 8 (Fig. 3a).

As differences between pre-perturbation and post-perturbation steptimes indicate a degree of ease or "fluidity of motion" with which animals learned to avoid the obstacle, we tested the temporal extent of the difference between pre-perturbation and post-perturbation steptimes for the Hx group. We hypothesized that lengthening the challenge session would promote learning, eventually resulting in little difference between pre-perturbation and post-perturbation steptimes within the Hx group. Therefore, we designed a ladder paradigm with four additional challenge sessions, which we refer to as the "extended challenge paradigm" depicted as "Paradigm 2" in Fig. 1a. Contrary to our prediction, we observed that, in the extended challenge sessions, Hx animals still displayed significant differences between mean pre-perturbation and post-perturbation steptimes, which did not resolve even on session 12 (Fig. 3c). During session 8, in the Nx group, mean post-perturbation steptime was $194 \pm 22.7$ ms, and pre-perturbation steptime was $141.4 \pm 8.6$ ms, yielding a

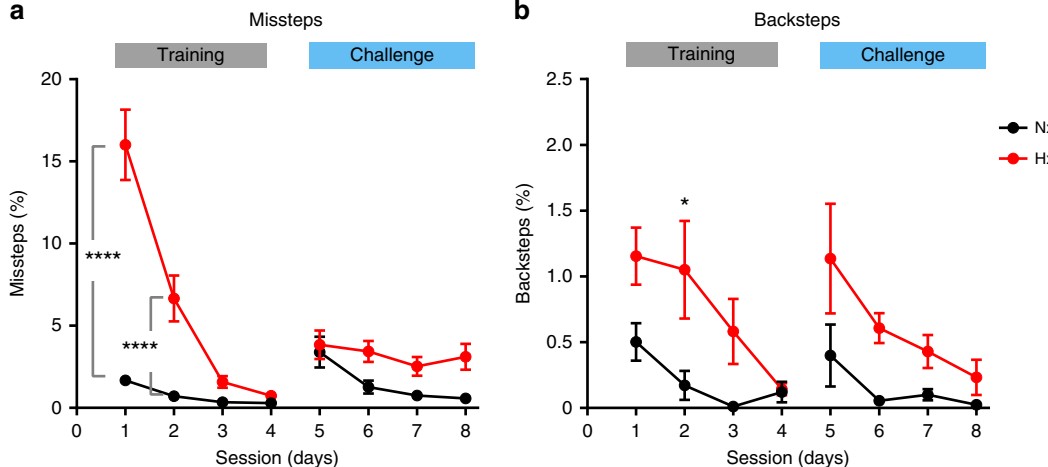

**Fig. 2** Locomotor performance measured using the Erasmus Ladder is altered in Hx mice. **a** Misstep percentage recorded by the Erasmus Ladder for Nx (black, $n = 8$), and Hx mice (red, $n = 11$). Training sessions (day 1 through day 4) are highlighted by overhanging grey bar ("training"). Challenge sessions (day 5 through day 8) are highlighted by overhanging blue bar ("challenge"). Statistics: two-way ANOVA, treatment effect $F_{(1,136)} = 63.08$, $P < 0.0001$, session effect $F_{(7,136)} = 16.85$, $P < 0.0001$, treatment × session effect $F_{(7,136)} = 13.17$, $P < 0.0001$. **b** Backstep percentage recorded during the same sessions. Statistics: two-way ANOVA, treatment × session effect $F_{(7,136)} = 0.9$, $P = 0.5$, treatment effect $F_{(7,136)} = 3.39$, $P = 0.0023$, session effect $F_{(7,136)} = 21.53$, $P < 0.0001$. Error bars denote SEM. Note difference in $y$ scale between panels. Asterisks represent results of Sidak's multiple comparison test following two-way ANOVA. ****$P < 0.0001$, *$P < 0.05$

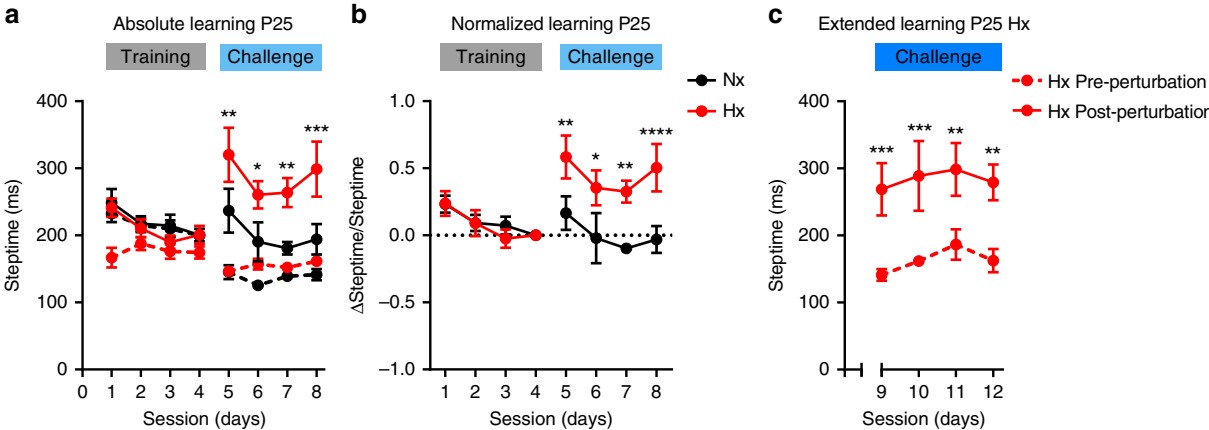

**Fig. 3** A conditioned learning paradigm using the Erasmus Ladder reveals adaptive cerebellar learning deficits in P25 Hx mice. **a** Absolute learning in P25 mice measured as a function of post-perturbation steptimes in Nx (black, solid) and Hx (red, solid) mice. Pre-perturbation steptimes (dashed traces) represent internal controls for Nx (black) and Hx (red). Nx: $n = 8$ animals, Hx: $n = 11$ animals. Statistics: two-way repeated measures (RM) ANOVA, matching: stacked along sessions, treatment × session effect $F_{(21,238)} = 5.152$, $P < 0.0001$, session effect $F_{(7,238)} = 2.678$, $P = 0.011$, treatment effect $F_{(3,34)} = 16.52$ 21.53, $P < 0.0001$. **b** Normalized learning in P25 mice quantified by normalizing post-perturbation steptimes for each mouse to post-perturbation steptime during session 4 for that mouse. Dotted line at $y = 0$ demarcates worse adaptive learning ($\Delta$Steptime/Steptime $> 0$) from better adaptive learning ($\Delta$Steptime/Steptime $< 0$). Statistics: two-way repeated measures (RM) ANOVA, matching: stacked along sessions, treatment × session effect $F_{(21,238)} = 5.185$, $P < 0.0001$, session effect $F_{(7,238)} = 2.625$, $P = 0.0125$, treatment effect $F_{(3,34)} = 17.75$, $P < 0.0001$. **c** Extended learning using paradigm 3 in P25 Hx mice as a function of pre-perturbation (red, dashed) and post-perturbation steptimes (red, solid) for session 9 through 12. Statistics: two-way RM-ANOVA, matching: both factors, perturbation effect $F_{(1,2)} = 21.67$, $P = 0.0432$, session effect $F_{(3,6)} = 2.078$, $P = 0.2046$, perturbation × session effect $F_{(3,6)} = 0.2315$, $P = 0.8713$, Sidak's multiple comparison test: session 9, $P = 0.0009$, session 10, $P = 0.0009$, session 11, $P = 0.0018$, session 12, $P = 0.0014$. Error bars denote SEM. For **a** and **b**, asterisks represent results of Tukey's multiple comparison test following two-way RM-ANOVA. *$P < 0.05$, **$P < 0.01$, ***$P < 0.001$, ****$P < 0.0001$. For **c**, asterisks represent result of Sidak's multiple comparison test following two-way RM-ANOVA. **$P < 0.01$, ***$P < 0.001$

difference of $52.6 \pm 14$ ms. By comparison, in the Hx group during session 12, mean post-perturbation steptime is $279.2 \pm 26.8$ ms and mean pre-perturbation steptime is $162.6 \pm 17.5$ ms, which yields a much higher steptime difference of $116 \pm 32$ ms. Also noteworthy is that the average post-perturbation steptime even in session 12 for Hx mice was above the 250 ms threshold, whereas normal mice—on average—adapt their movements to avoid the obstacle within the threshold by session 6 ($190.5 \pm 29$ ms). In summary, chronic perinatal Hx results in significant

adaptive cerebellar learning deficits, which are not resolved even after extended challenge to promote learning.

**Adaptive cerebellar learning deficits are long lasting**. As neonatal brain injury has long-term consequences on locomotor, cognitive, and behavioral outcomes[6,34], we tested the effect of chronic perinatal Hx on adaptive cerebellar learning in naive mice, 4.9 weeks after hypoxic rearing (Fig. 1a, Paradigm 3). On the first day of challenge, P45 Nx mice had a mean post-

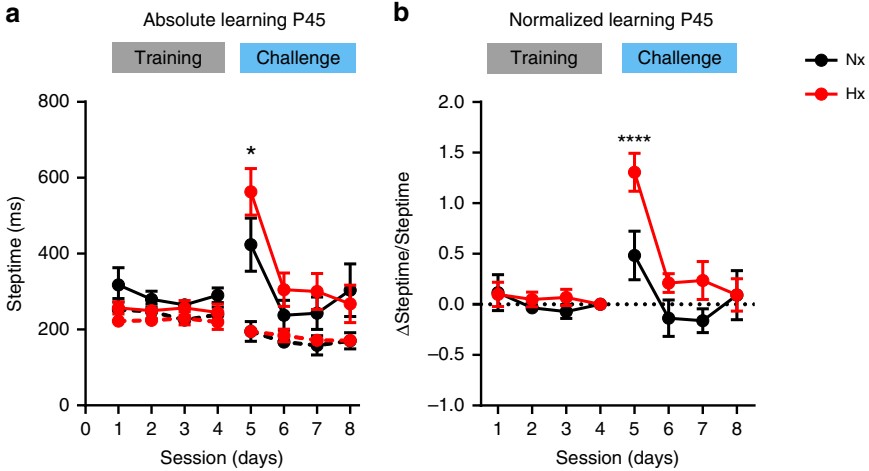

**Fig. 4** Adaptive cerebellar learning deficits in P45 Hx mice. **a** Absolute learning in P45 mice measured as a function of post-perturbation steptimes in Nx (black, solid) and Hx (red, solid) mice. Pre-perturbation steptimes (dashed traces) represent internal controls for Nx (black) and Hx (red). Nx: $n = 7$ animals, Hx: $n = 7$ animals. Statistics: two-way RM-ANOVA, matching: stacked along sessions, session effect $F_{(7,168)} = 8.706$, $P < 0.0001$, treatment effect $F_{(3,24)} = 7.502$, $P = 0.001$, treatment × session effect $F_{(21,168)} = 4.766$, $P < 0.0001$. **b** Normalized learning in P45 mice quantified by normalizing post-perturbation steptimes for each mouse to post-perturbation steptime during session 4 for that mouse. Dotted line at $y = 0$ demarcates worse adaptive learning ($\Delta$Steptime/Steptime $> 0$) from better adaptive learning ($\Delta$Steptime/Steptime $< 0$). Statistics: two-way RM-ANOVA, matching: stacked along sessions, session effect $F_{(7,168)} = 9.552$, $P < 0.0001$, treatment effect $F_{(3,24)} = 7.675$, $P = 0.0009$, treatment × session effect $F_{(21,168)} = 5.431$, $P < 0.0001$. Error bars denote SEM. Asterisks represent results of Tukey's multiple comparison test following two-way ANOVA, *$P < 0.05$, **$P < 0.01$, ***$P < 0.001$, ****$P < 0.0001$

perturbation steptime of $423.7 \pm 70.3$, however Hx mice displayed a higher post-perturbation steptime of $563.2 \pm 61.5$, indicating the presence of adaptive cerebellar learning deficit for the first challenge session (Fig. 4a). This deficit was even more obvious when normalized learning was compared between Hx and Nx (Fig. 4b). Contrasted to the P25 Hx group, P45 Hx animals showed lower differences in post-perturbation steptimes compared with age-matched Nx animals. Multiple comparison hypothesis tests indicated no significant differences between post-perturbation steptimes of Hx and Nx groups in subsequent sessions using both absolute and normalized metrics. In conclusion, although older Hx animals eventually display cerebellar learning on par with Nx animals in later sessions, significant adaptive locomotion deficits were still observed in the Hx group.

**Neonatal brain injury induces altered stepping patterns.** Cerebellar damage or dysfunction owing to injury or genetic causes has been correlated to abnormalities in multi-joint limb control and intra-limb coordination, resulting in changes to gait pattern[32,35]. We thus sought to determine changes in stepping patterns in Hx animals using the ladder's step classification system (Fig. 1b, "Short steps"). Alterations in stride length are typical of mice with cerebellar deficits in gait[36,37]. Reduced stride length is a compensatory mechanism, which decreases the likelihood of falls in humans[38,39]. So, we performed start-point and end-point analysis, specifically of short steps during training and challenge sessions for both P25 and P45 groups (Fig. 5). Our analysis revealed an altered locomotor-stepping pattern in Hx mice across age, with some key similarities to Nx animals. First, we observed that basal stepping patterns were inversely related at the start of the training session; in Nx animals, there was an increase in the percentage of short steps when comparing P25 with P45 groups (Fig. 5a). Conversely, for Hx animals, there is a decrease in percentage of short steps comparing P25 with P45 groups. Second, by the end of training, both Nx and Hx P25 groups displayed strikingly similar percentages of short steps (Fig. 5b). However, in both Nx and Hx P45 groups, short step patterns do not appear to be differentially changed after training, even though Hx animals

exhibit a higher percentage of short steps in both sessions (Fig. 5a, b).

During the challenge sessions, both P25 and P45 Hx groups displayed a decrease in short step percentage, although in Nx groups at both ages, short steps either mildly increased or were relatively similar (Fig. 5c, d). Thus, locomotor-stepping patterns in younger Hx animals displayed a trend toward normal behavior over training, however, this gradually reverted by the end of challenge sessions. In older Hx animals, stepping patterns during training sessions stayed relatively the same (~ 80%). During the challenge sessions, short step percentages in older Hx animals were reduced to < 60%, but remained significantly higher than the older Nx group (Fig. 5c, d).

**Neonatal brain injury disrupts spontaneous PC firing.** As we observed significant deficits in locomotor performance and adaptive cerebellar learning, we sought to determine a potential physiological basis for behavioral abnormalities in the cerebellar cortex of Hx animals. Previously, we had observed abnormal synaptic activity of GABAergic interneurons in cerebellar white matter of mice following Hx[29]. From a behavioral standpoint, however, dysfunctional PCs, rather than white matter interneurons, constitute a more direct source of adaptive locomotor dysfunction. As PCs are the sole output of the cerebellar cortex, dysfunction in PC firing can be a critical determinant of abnormal locomotor behavior. This is adequately clear from numerous studies where both genetic and injury-related locomotor deficits stem from abnormal PC function or development[16,40].

In our previous study, we also demonstrated that PCs have a significantly lower firing rate after Hx[29]. However, these recordings were performed with cerebellar slices. We thus decided to confirm our earlier findings in PCs and extend our studies into later ages following Hx, using a more intact, in vivo methodology. Importantly, as PCs play an important role in the cerebellar learning circuitry, we also sought to define potential changes in evoked firing following Hx. Therefore, we decided to first identify the modulation and plasticity of PC firing frequency over the course of Nx postnatal development using our in vivo

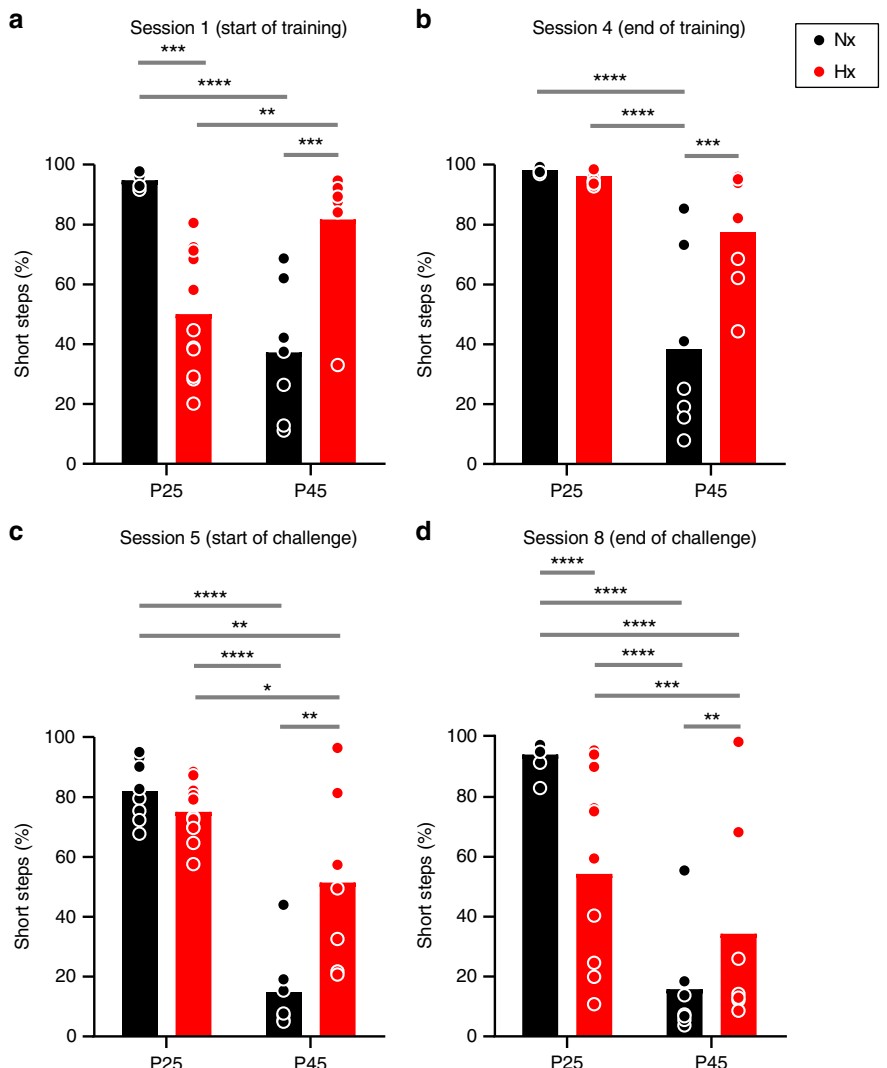

**Fig. 5** Locomotor-stepping pattern is altered in Hx mice across age groups. **a** Short step percentage measurement in session 1 (start of training) in P25 and P45 Nx (black bar graphs, P25 Nx: $n = 8$ animals, P45 Nx, $n = 7$ animals), and Hx, respectively, (red bar graphs, P25 Hx: $n = 11$ animals, P45 Hx, $n = 7$ animals). Statistics: two-way ANOVA, age × treatment effect $F_{(1,29)} = 44.81$, $P < 0.0001$, age effect $F_{(1,29)} = 3.775$, $P = 0.0618$, treatment effect $F_{(1,29)} = 0.0007305$, $P = 0.9786$. **b** Short steps in session 4 (end of training). Statistics: age × treatment effect $F_{(1,29)} = 12.67$, $P = 0.0013$, age effect $F_{(1,29)} = 45.6$, $P < 0.0001$, treatment effect $F_{(1,29)} = 10.17$, $P = 0.0034$. **c** Short steps in session 5 (start of challenge), statistics: age × treatment effect $F_{(1,29)} = 13.89$, $P = 0.0008$, age effect $F_{(1,29)} = 60.34$, $P < 0.0001$, treatment effect $F_{(1,29)} = 6.405$, $P = 0.0171$. **d** Session 8 (end of challenge). Statistics: age × treatment effect $F_{(1,29)} = 82.15$, $P < 0.0001$, age effect $F_{(1,29)} = 232.0$, $P < 0.0001$, treatment effect $F_{(1,29)} = 11.08$, $P = 0.0024$. Error bars denote SEM. Asterisks represent results of Tukey's multiple comparison test following two-way ANOVA *$P < 0.05$, **$P < 0.01$, ***$P < 0.001$, ****$P < 0.0001$

setup. We acquired in vivo multielectrode array (MEA) recordings of PCs at P13 and P21 (Fig. 6a, b) from cerebellar cortices of *Pcp2-Cre* mice (Mpin line) previously injected with Cre-dependent Channelrhodopsin-mCherry (ChR2-mCherry) adenovirus (see Methods). We calculated the simple spike firing frequency of PCs for both groups, and analyzed spontaneous activity as well as evoked (optostimulated) activity over a 2 s total time period before and after optostimulation onset ($t = 0$) shown in Fig. 6c, d. Optostimulation was presented for a total of 1000 ms (Fig. 6a–d, blue lines). We compared mean basal firing frequencies in a 2 s time window (1 s pre-optostimulation onset and 1 s post-optostimulation onset; "optostimulation" onset implies $t = 0$ in firing frequency vs. time plots) between Nx and Hx across ages P13 and P21 and found statistically significant differences across groups (Fig. 6e). Results for changes in evoked firing will be detailed in the following sub-section. Comparing between ages P13 (black) and P21 (blue) in Nx animals, PCs

displayed an increase in spontaneous/basal firing frequency. The mean basal firing frequency in PCs increased from 27.83 Hz at P13 (Fig. 6e, black open circles) to 55.09 Hz at P21 (blue open circles) under Nx conditions (Fig. 6e). In comparison, following Hx, PCs at P13 (red open circles) as well as P21 (wine open circles) displayed severely reduced basal firing frequency compared with PCs in age-matched Nx controls. This suggested that the developmental time from P13 to P21[41] following Hx was not sufficient to endogenously recover spontaneous activity to levels comparable to firing frequencies in age-matched Nx animals. Further, there were no statistically significant changes in basal firing between P13 and P21 Hx groups (Fig. 6e).

In addition to basal firing frequency, we also analyzed dynamics of basal PC firing by comparing CV and CV2 values between groups (Fig. 6f, g). We found statistically significant differences across groups for CV values, but not for CV2 values ($n = 15–37$ cells per group from 8 to 10 animals per group).

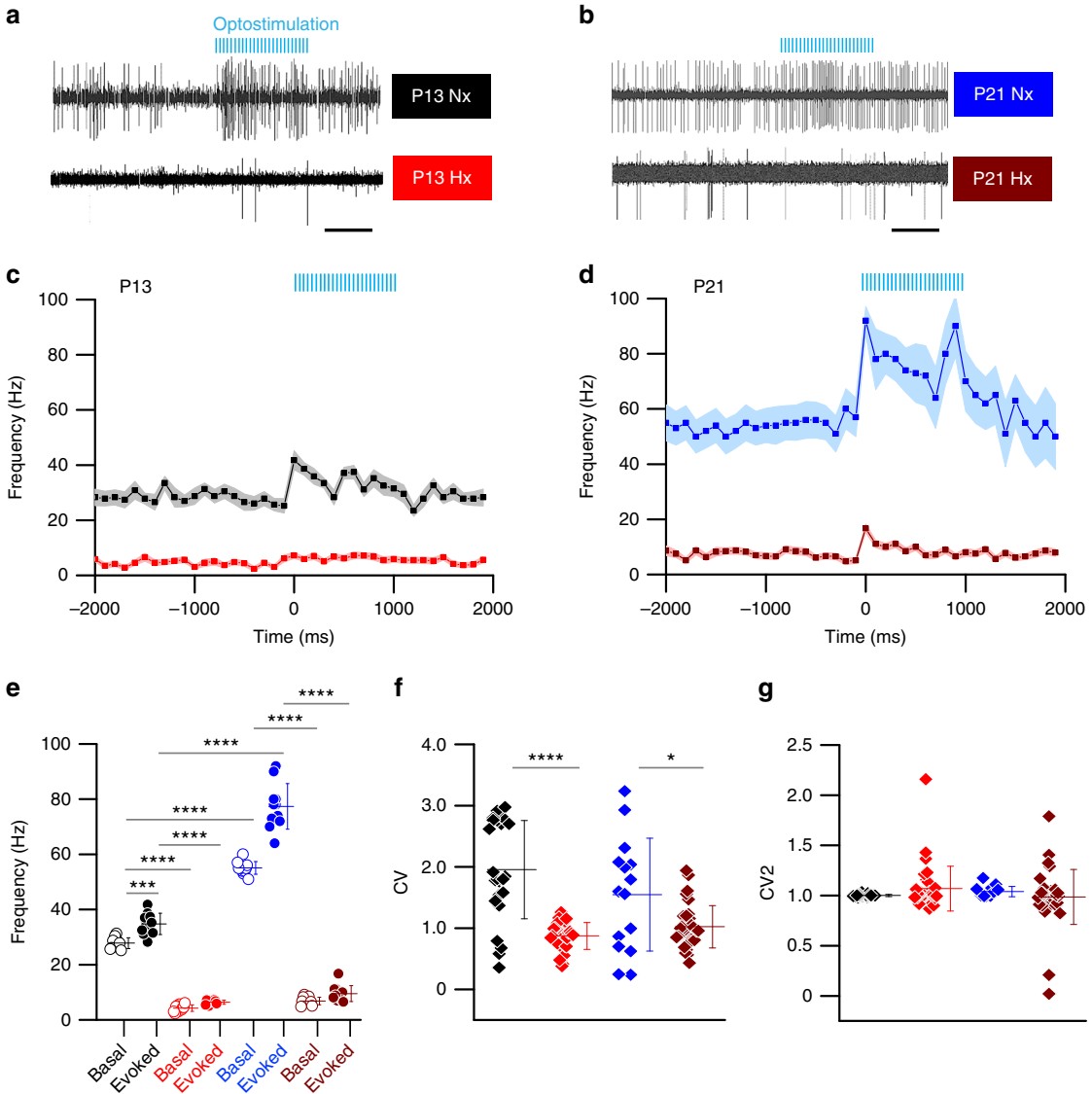

**Fig. 6** Altered in vivo Purkinje cell simple spike firing rates in Hx and Nx *Pcp2-Cre* mice at P13 and P21. **a** Representative multielectrode array (MEA) recordings from Purkinje cells (PCs) in Nx (black) and Hx (red) animals at P13. Blue bars represent optostimulation. **b** Representative MEA recordings from PCs in Nx (blue) and Hx (wine) animals at P21. **c** Mean simple spike (SS) firing frequency of PCs in Nx P13 (black, $n = 25$ PCs), and Hx P13 animals (red, $n = 37$ PCs); shaded area indicates SEM for P13 Nx (light grey), and P13 Hx (light red). **d** Mean SS frequency of PCs in Nx P21 (blue, $n = 15$ PCs) and Hx P21 (wine, $n = 33$ PCs). SEM indicated for P21 Nx (light blue) and P21 Hx (light red). Data obtained from multiple PCs from 8–10 animals/group for **c–d**. **e** Scatter plot SS frequency comparison of Nx P13 basal (black, open), Nx P13 evoked (black, solid), Hx P13 basal (red, open), Hx P13 evoked (red, solid), Nx P21 basal (blue, open), Nx P21 evoked (blue, solid), Hx P21 basal (wine, open), and Hx P21 evoked (wine, closed). Each circle represents mean frequency obtained from multiple PCs/group for one sample. Ten successive timepoints were sampled pre-optostimulation and 11 successive timepoints were sampled post-optostimulation. Horizontal lines in each group represent the mean firing frequency across time-samples. One-way ANOVA, $F_{(7,80)} = 607.3$, $P < 0.0001$, $R^2 = 0.9815$. **f** Scatter plot showing coefficient of variance of interspike interval (CV) for basal PC firing in Nx and Hx animals. Each diamond represents individual CV values from the same set of PCs for which frequency plots are analyzed in **c–e**. One-way ANOVA, $F_{(3,106)} = 22.24$, $P < 0.0001$, $R^2 = 0.3863$. **g** Scatter plot showing CV2 values of basal PC firing in Nx and Hx animals from the same data set as **c–f**. One-way ANOVA, $F_{(3,108)} = 1.144$, $P = 0.3349$, $R^2 = 0.03079$). Scale in **a** and **b** represent 500 ms. Error bars in **e–g** denote SEM. Asterisks represent results of Tukey's multiple comparison test following one-way ANOVA *$P < 0.05$, ***$P < 0.001$, ****$P < 0.0001$

Although basal firing frequency increased with maturity, comparing between groups, we found that the mean CV value did not change between P13 Nx and P21 Nx. However, we found that mean CV was much lower in the P13 Hx group. A lower mean CV was also found in P21 Hx group compared with age-matched Nx control. Furthermore, spike profile analysis also showed differences in distribution, indicating a delayed profile in Hx groups (Supplementary fig. 6a–d, see below). Thus, Hx induces a reduction in basal firing frequency as well as altered CV values, and spike profiles over the course of postnatal development.

**PC evoked activity is altered after neonatal brain injury.** Our behavioral results using the Erasmus ladder clearly indicate abnormal steptimes in Hx mice (Figs. 3 and 4). PC excitability has been implicated in the temporal aspects of associative motor learning[42,43]. Importantly, emerging evidence strongly indicates

that activity of PCs per se–irrespective of climbing fiber (CF) input—can drive motor skill learning[44]. Thus, we sought to determine neuronal excitability via evoked PC firing rates in Hx and Nx animals using channelrhodopsin-mediated optostimulation. We developed and optimized stereotaxic channelrhodopsin-mCherry (AAV9-DIO-ChR2-mCherry, Supplementary fig. 1a) AAV injections to the cerebellar cortex of juvenile *Pcp2-Cre* mice (P9) obtaining robust ChR2-mCherry expression within 4 days post injection (Supplementary fig. 1b, c). To ensure viability and responsiveness of PCs following neonatal Hx treatment and ChR2-mCherry AAV injection, we performed in vitro slice electrophysiology, where we analyzed passive properties (Supplementary Table 1) and response to blue light stimulation (Supplementary fig. 2). Our results show that Hx PCs are electrophysiologically viable (Supplementary fig. 2a–f), do not show altered passive properties (Supplementary Table 1), and respond to optostimulation is a manner similar to Nx PCs (Supplementary fig. 2g–j). Finally, we also morphologically (Supplementary fig. 3) and functionally verified the cellular specificity of viral expression, and photoresponse to optogenetic stimulation in the cerebellar cortex of Pcp2-Cre Mpin mice. In order to address potential non-specificity in photoresponse in our experiments (all of which involved AAV injections into a specific region of the cerebellar cortex of Pcp2-Cre Mpin mice), we recorded from randomly selected molecular layer interneurons in the vicinity of mCherry+ PCs, whereas stimulating with blue light. Compared with the direct and robust functional response of mCherry+ PCs to optostimulation (Supplementary fig. 4a), we observed that none of the molecular layer interneurons (MLIs) we recorded from had a photocurrent when optostimulation was applied (0/19 MLIs from five mice; Supplementary fig. 4b). We ensured viability of MLIs to fire action potentials by using a current injection protocol (Supplementary fig. 4c). Thus, although we observed rare instances of mCherry+ MLIs, the lack of photocurrent following optostimulation from MLIs in the vicinity of the mCherry+ PCs suggests that our viral strategy and optostimulation paradigm is functionally selective for PC responses.

Our in vivo firing frequency analysis was performed in a defined time window of optostimulation, ranging a total of 1000 ms, indicated by the light blue bars depicted in Fig. 6a–d. We found that evoked activity of PCs in both P13 Nx and P21 Nx groups were significantly higher than their respective basal firing

rates (Fig. 6e). We also observed that evoked activity of PCs in P21 Nx animals displayed significantly higher firing frequencies than P13 Nx animals (Fig. 6e). Surprisingly, however, Hx PCs failed to evoke even moderately high spike activity with the same optostimulation paradigm (Fig. 6a–e), despite robust ChR2-mCherry fusion protein expression in PCs from virus-injected P13 Hx and P21 Hx mice (Supplementary fig. 5a). In vivo evoked firing frequency for both P13 and P21 Hx groups satisfied our cell selection criteria (average evoked firing frequency for 1000 ms after the first stimulus > average basal firing frequency for 1000 ms before the first stimulus + 1 S.D.) (Supplementary fig. 5b, c). Finally, we confirmed that the electrode position was at the PC soma layer (Supplementary fig. 5, insets in b and c).

We found that the increase in evoked firing for Hx mice groups (both ages) were significantly weaker than age-matched normoxia controls with the same optostimulation paradigm. We also found no statistically significant differences between evoked and basal firing rates within Hx animals in both age groups (Fig. 6e). The mean firing frequency shifted from 4.3 Hz (basal) to 6.43 Hz (optostimulated) and 6.85 Hz (basal) to 9.55 Hz (optostimulated) for Hx P13 and Hx P21, respectively, suggesting a trend toward delayed functional recovery in PCs after Hx.

**A GABA reuptake inhibitor rescues motor and PC abnormalities**. Previously, we reported that increasing levels of neurotransmitter GABA immediately following chronic perinatal Hx rescues cellular changes in OPC differentiation and GABAergic interneuron cell number and physiology in the cerebellar white matter[29]. However, whether pharmacological intervention with GABA reuptake inhibitors translated into an observable effect on behavior was unexplored. Here, we used a similar intervention paradigm, where we administered the GABA reuptake inhibitor, Tiagabine i.p. once per day from P12 through P16 (Fig. 1a). We then used the ladder paradigm to test the effect of Tiagabine on motor performance 2 weeks after hypoxic rearing (Fig. 7). Overall, there were clear intervention effects as well as session effects between Tiagabine-treated and control animals. We observed a slight statistically insignificant decrease in misstep percentage in the Tiagabine-treated group (Fig. 7a, green trace) during the first session. However, during session 2, there was a dramatic and statistically significant decrease in percentage of missteps in the treated group, which made 2.6 ± 0.85% missteps,

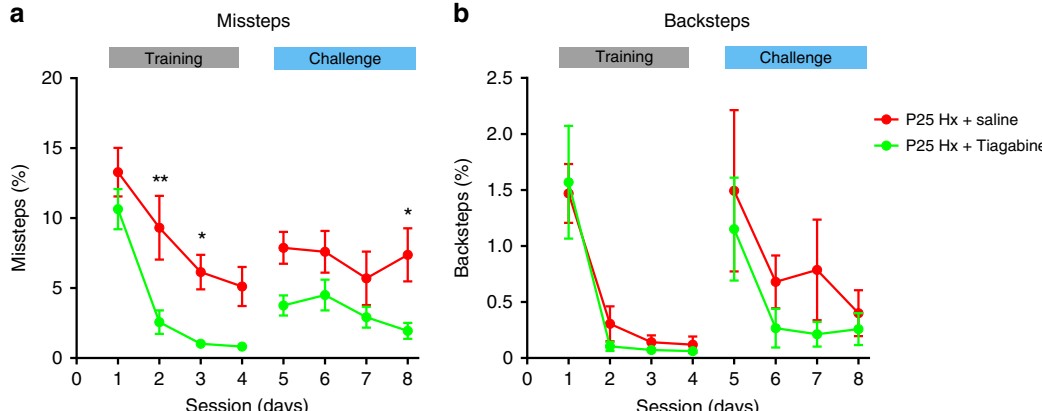

**Fig. 7** Pharmacological intervention using a GABA reuptake inhibitor ameliorates locomotor malperformance in Hx mice. **a** Comparison of misstep percentages between Tiagabine-treated Hx animals (green trace, P25 Hx + Tiagabine, n = 6 animals) and controls (red trace, P25 Hx + saline, n = 5 animals). Statistics: two-way ANOVA, intervention effect $F_{(1,72)} = 46.91$, $P < 0.0001$, session effect $F_{(7,72)} = 9.957$, $P < 0.0001$, intervention × session $F_{(7,72)} = 0.6637$, $P = 0.7018$. **b** Comparison of backstep percentages between Tiagabine-treated (green trace) and controls (red trace). Statistics: two-way ANOVA, intervention effect $F_{(1,72)} = 2.064$, $P = 0.1551$, session effect $F_{(7,72)} = 6.903$, $P < 0.0001$, intervention × session $F_{(7,72)} = 0.2729$, $P = 0.9625$. Error bars denote SEM. Note difference in *y* scale between panels. Asterisks represent results of Sidak's multiple comparison test following two-way ANOVA. ** $P < 0.01$, * $P < 0.05$

compared with saline Hx group which made 9.3 ± 2.28% missteps (Fig. 7a). On the following session 3, this downward trend in missteps continued in the Tiagabine-treated animals, which made only 1 ± 0.16% missteps. During the challenge sessions 5 through 7, misstep percentages were still lower in the treated group, although not statistically significant (Fig. 7a). However, on the final challenge session, there was a statistically significant difference in misstep percentages, in which the treated animals performed better compared with the saline treated animals. Although we observed an overall session effect, our internal control measurements of backsteps did not show any intervention effects between treated and control groups. Multiple comparison tests showed that differences in backsteps were not statistically significant in any of the sessions between treated and untreated Hx groups (Fig. 7b), suggesting a locomotor-specific improvement in the treated group. Finally, we also monitored the effect of Tiagabine treatment on adaptive cerebellar learning (Supplementary fig. 7). Interestingly, we did not observe any improvement in cerebellar learning in either P25- or P45-treated groups compared with saline controls. Consistent with behavioral observations at P45, in vivo electrophysiology at P45 indicated that Hx causes a long-term reduction in PC simple spike firing patterns (Supplementary fig. 8). In conclusion, we observed a dramatic rate of improvement in motor performance in Hx animals after treatment with Tiagabine, but not in adaptive cerebellar learning.

In order to correlate the behavioral rescue in locomotor performance following Tiagabine intervention in Hx animals to pathophysiology, we used in vivo optogenetics and electrophysiology as described in detail in previous sections. We decided to perform electrophysiological analysis in P30 animals, as this time point lies within the time period analyzed for younger Hx animals (P25–P33) on the Erasmus Ladder, where we observed drastic effects on cerebellar learning and locomotor performance. We found that Tiagabine treatment partially improves some characteristics of PC firing. We first observed that PCs in the P30 Hx group have significantly reduced basal and evoked firing frequency compared with Nx 30 PCs (Fig. 8c, compare black open and solid circles to red open and solid circles, respectively). However, basal as well as evoked PC firing frequency were significantly improved in the Tiagabine-treated P30 Hx group, compared with the untreated Hx group (Fig. 8c, compare red open and solid circles with green open and solid circles). Analysis showed that there were statistically significant differences in both basal and evoked firing frequency between the Tiagabine-treated Hx group and the untreated Nx group (Fig. 8c, compare green open and solid circles to light brown open and solid circles, respectively), indicating that our paradigm of pharmacological intervention partially rescued PC firing frequency. We also noted that Tiagabine treatment may have an effect on PC firing in Nx animals (Fig. 8a–c). This effect was limited to basal PC firing and did not affect evoked firing frequency.

Finally, we found that there was an overall statistically significant difference across treated and untreated Hx and Nx groups when comparing mean CV values (Fig. 8d), but inter-group differences in CV were not significant. We noted both across group and inter-group differences in CV2 values (Fig. 8e). The untreated Hx group showed statistically significant differences in CV2 compared with the untreated Nx group and the Tiagabine-treated Nx group. However, CV2 for the Tiagabine-treated Hx group was not statistically significant compared with either Nx untreated or the Tiagabine-treated Nx group. This suggested to us that there was a trend toward improvement in rhythmicity of PC firing following Tiagabine treatment in Hx mice. This trend was also observed in spike profile analysis, which indicated an improved profile in the Tiagabine-treated group (Supplementary figure 6e–h). In summary, results from Erasmus

Ladder data and in vivo electrophysiology indicate that Tiagabine partially rescues motor malperformance potentially due to an improvement in specific PC firing characteristics such as basal simple spike firing frequency, rhythmicity of firing, and mean simple spike duration.

**Tiagabine restores evoked PC spike firing after injury**. Electrophysiological recordings from PCs show that complex spikes are mainly responsible for the adaptive conditioned response in cerebellar learning[45]. As the Erasmus Ladder uses a conditioned learning paradigm, we analyzed complex spike patterns, which may potentially underlie behavioral deficits we observed. We sorted PC complex spikes from P30 Nx and Hx firing data (Fig. 9a–c) and analyzed basal and evoked complex spike firing patterns in treated and untreated Nx and Hx groups (Fig. 9d, e). Median latency to first complex spike was 139 ms, compared with simple spike latency, which was 6 ms (Supplementary fig. 9). We found an overall statistical difference in mean basal and evoked firing frequency between treated and untreated groups (Fig. 9f). Comparing Nx and Hx untreated groups, we report a statistically significant inter-group difference in mean evoked complex spike firing frequency. Surprisingly, however, in contrast to our observations of simple spike firing between groups, we did not observe a significant difference in mean basal complex spike firing frequency between Nx and Hx untreated groups. Tiagabine treatment did in fact rescue mean evoked complex spike firing frequency to near-normal levels. Although we did not observe significant inter-group differences between mean basal complex spike firing frequency in treated and untreated Hx and Nx groups, we did observe differences in CV values or regularity in basal complex spike firing. Higher mean CV values are obtained in Hx group compared with the Nx group (Fig. 9g). This increase in mean CV is restored to near-normal levels in the Tiagabine-treated Hx group. We did not observe an alteration in rhythmicity of basal complex spike firing across groups (Fig. 9h). Taken together, our complex spike analysis indicates an alteration in the generation of evoked complex spike firing frequency and regularity, which is rescued to near-normal levels in Hx animals treated with Tiagabine.

## Discussion

We have demonstrated the presence of locomotor malperformance, cerebellar learning deficits, and abnormal in vivo PC physiology in a clinically relevant mouse model of neonatal brain injury. Our study involves the rigorous and objective analysis of locomotor behavior and cerebellar learning in Hx-induced neonatal brain injury. Using the advanced Erasmus Ladder behavioral system, we have shown that Hx animals present with locomotor miscoordination, long-term cerebellar learning deficits, and altered locomotor-stepping patterns. Our study also employs in vivo multielectrode recording and optogenetics to define critical features of the pathophysiology of neonatal brain injury. We found that PCs in Hx mice have significantly lower simple spike firing frequencies in both basal and evoked conditions across different developmental timepoints. Finally, we have provided evidence for the pharmacological efficacy of targeting the developmental GABA system in ameliorating Hx-induced locomotor performance deficits. Treatment with the GABA reuptake inhibitor Tiagabine immediately following Hx resulted in a significant reduction in the percentage of misstep compared with controls. Hx animals treated with Tiagabine also displayed improved PC simple spike and complex spike firing patterns compared with untreated animals. Findings from the present study provide a potential mechanistic basis for linking specific alterations in cerebellar behavior to cellular and physiological

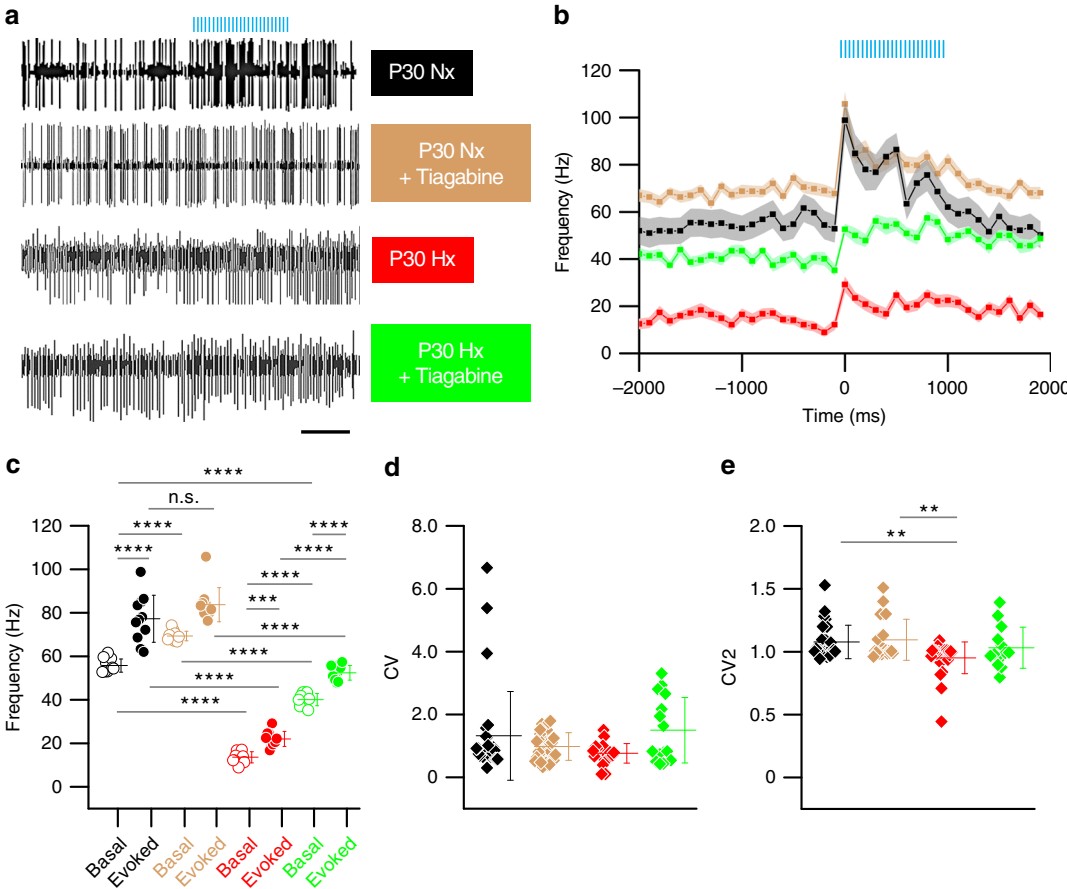

**Fig. 8** Pharmacological intervention improves PC simple spike firing frequency in Hx *Pcp2-Cre* mice in vivo. **a** Representative MEA recordings from PCs in untreated P30 Nx (black), Tiagabine-treated P30 Nx (light brown), untreated P30 Hx (red), and Tiagabine-treated P30 Hx (green) animals. Scale 500 ms. **b** Mean SS firing frequency in animals from untreated P30 Nx group (black, $n = 31$ PCs, SEM—light grey), Tiagabine-treated Nx group (light brown, $n = 37$ PCs SEM—light tan), untreated Hx group (red, $n = 25$ PCs, SEM—light red), and Tiagabine-treated Hx group (green, $n = 18$ PCs, SEM—light green) obtained from 8–10 animals per group. **c** Scatter plot SS firing frequency comparison across groups as indicated by legend in **a**. For all groups, open circles represent basal firing and solid circles represent evoked (optostimulated) firing. Each circle represents mean firing frequency obtained from multiple cells per group for a single sample. Ten successive timepoints were sampled pre-optostimulation and 11 successive timepoints were sampled post-optostimulation. Horizontal lines in each group represent the mean firing frequency across time-samples. One-way ANOVA, $F_{(7,80)} = 244.7$, $P < 0.0001$, $R^2 = 0.9554$. **d** Scatter plot showing coefficient of variance of interspike interval (CV) values of basal PC firing in Nx and Hx animals. Each diamond represents individual CV values from the same set of PCs for which frequency plots are analyzed in **b–c**. (One-way ANOVA, $F_{(3,103)} = 2.109$, $P = 0.0377$, $R^2 = 0.07836$). **e** Scatter plot showing CV2 values of basal PC firing in Nx and Hx animals from the same data set as in **b–d**. (One-way ANOVA, $F_{(3,89)} = 5.18$, $P = 0.0024$, $R^2 = 0.1487$). Scale in **a** and **b** represent 500 ms. Error bars in **b–e** denote SEM. Asterisks represent results of Tukey's multiple comparison test following one-way ANOVA. **$P < 0.01$, ***$P < 0.001$, ****$P < 0.0001$

changes caused by neonatal brain injury. Thus, our combined bidirectional approach of using automated quantification of cerebellar-mediated behaviors and in vivo optogenetic–electrophysiological analysis provides a pre-clinical platform for functional assessment of targeted therapeutics for prematurity-related movement disorders.

Proper development of the cerebellar cortex ensures precise spatiotemporal control of sensorimotor behavior at later stages. Disruptions or delays in development owing to injury lead to abnormalities in motor behavior[16]. Our observation of locomotor malperformance in Hx mice finds significant parallels in motor deficits observed in humans born prematurely. A meta-analysis of 41 studies involving human subjects revealed a significant impairment in development of motor skills in children born very preterm ( < 32 weeks gestation)[46]. As such, cerebellar injury leads to poor outcomes in basal and adaptive locomotor assessment owing to gait impairment[31], inter-limb and intra-limb mis-coordination[35], and defective integration of sensory information[47,48]. Recent reports in animal models provide strong

evidence for dynamic, real-time control of stepping behavior by the cerebellum[49], which is altered upon selective manipulation of different cell types in the cerebellar cortex including PCs[50]. In this context, our observation that Hx mice commit a significantly higher percentage of missteps correlates well with these studies and further validates the clinical relevance of our model to studying locomotor disorders primarily affecting cerebellar processing.

Besides basal locomotor behaviors such as running or walking, data from human patients clearly show that cerebellar injury also significantly impairs adaptive motor learning[32,51], which is a core feature of cerebellar function[52]. In animal models, well characterized and widely used cerebellar learning paradigms include delay eyeblink conditioning[53] and the vestibulo-ocular reflex paradigm[54]. However, these paradigms are restrictive, and do not involve whole-body locomotion. Our study directly addresses neurological sequelae of neonatal brain injury by identifying specific deficits in adaptive cerebellar learning within a locomotor context. Although it is possible that our learning paradigm is also

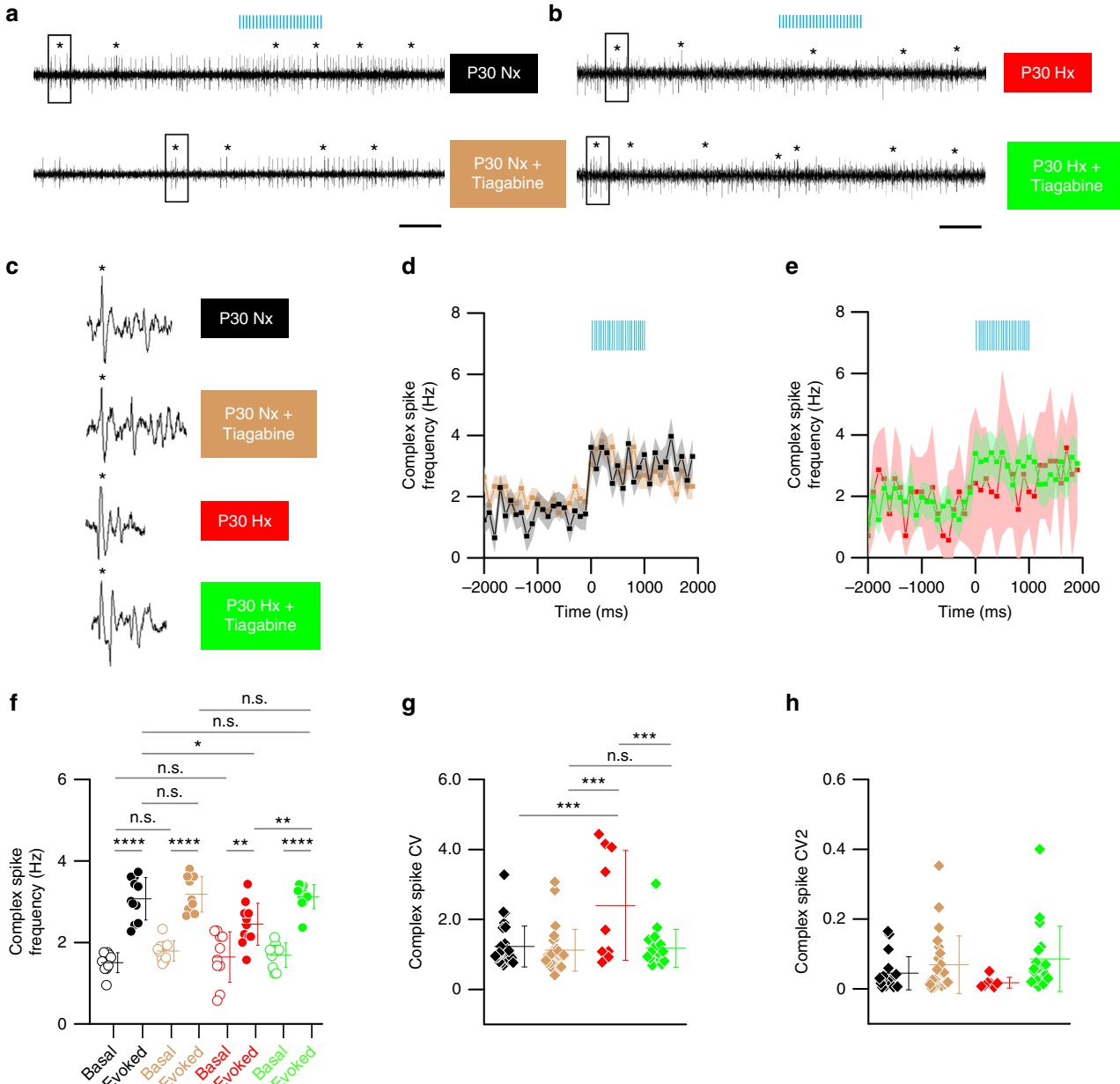

**Fig. 9** Pharmacological intervention restores complex spike firing to near-normal levels in Hx mice in vivo. **a** Representative MEA recordings from P30 Nx (black), P30 Nx treated (light brown), **b** P30 Hx untreated (red), P30 Hx treated (green) showing complex spikes (CSs) (Asterisks). Scale 500 ms. **c** Representative CSs corresponding to spikes within boxes in **a** shown with increased resolution. Asterisk marks primary spike of CS, which is followed by characteristic spikelets. **d** Mean CS frequency for P30 animals from untreated Nx (black, $n = 28$ PCs, SEM—light grey), Tiagabine-treated Nx (light brown, $n = 28$ PCs SEM—light tan), **e** untreated Hx (red, $n = 9$ PCs, SEM—light red), and Tiagabine-treated Hx (green, $n = 18$ PCs, SEM—light green) groups obtained from 8 to 10 animals/group. **f** Scatter plot CS frequency comparison across groups as indicated by legend in **a** and **b**. For all groups, open circles represent basal firing and solid circles represent evoked (optostimulated) firing. Each circle represents mean firing frequency obtained from multiple cells/group for one sample. Ten successive timepoints were sampled pre-optostimulation and 11 successive timepoints were sampled post-optostimulation. Horizontal lines in each group represent the mean firing frequency across time-samples. One-way ANOVA, $F_{(7,76)} = 31.86$, $P < 0.0001$, $R^2 = 0.7458$. **g** Scatter plot showing coefficient of variance of interspike interval (CV) values for basal PC firing in Nx and Hx. Each diamond represents individual CV values from the same set of PCs analyzed in **d**–**f**. (One-way ANOVA, $F_{(3,79)} = 7.378$, $P = 0.0002$, $R^2 = 0.2189$). **h** Scatter plot showing CV2 for basal PC firing in Nx and Hx animals from the same data set as in **d**–**f**. (One-way ANOVA, $F_{(3,66)} = 1.898$, $P = 0.1386$, $R^2 = 0.07941$). Scale in **a** and **b** represent 500 ms. Error bars in **b**–**e** denote SEM. Asterisks represent results of Tukey's multiple comparison test following one-way ANOVA, *$P < 0.05$, ***$P < 0.001$, ****$P < 0.0001$, n.s.—not significant

detecting extracerebellar modulations of the conditioned response (adapting steptimes for optimal obstacle avoidance), our experimental design minimizes the likelihood of contributions from regions other than cerebellum. Thus, our use of the Erasmus Ladder is well-suited to tease out differences in locomotor performance and adaptive cerebellar learning.

Although previous studies of cerebellar learning using the ladder have investigated only one age group for a given

experimental condition[30], or a broad age range within the adult stage[55], our study identifies specific changes in cerebellar learning across two defined age groups in normally developing mice to investigate developmental regulation of this function. We also show how these changes are subject to neurodevelopmental disruption in Hx mice. The timepoints analyzed—P25 and P45—approximately correspond to childhood and late adolescence/early young adulthood, respectively. Our data suggest that juvenile mice are quicker than young adults in adapting to a randomly placed obstacle in their path (Figs. 3a, 4a, session 5). However, the difference in adaptation-steptimes persists for more sessions, suggesting that motor adaptive deficits that we observe in Hx mice could possibly arise from a loss in cerebellar plasticity due to developmental delay[16]. That naive P45 mice can adapt their movements, whereas P25 mice cannot, is indicative of alternative locomotor strategies through compensatory circuit formation and/or delayed circuit maturation[56,57]. This is borne out in the stepping pattern data (Fig. 5), wherein P25 mice display a "catch-up" effect, where they "learn" to use higher percentage of short steps, similar to Nx mice (Fig. 5b, c). However, this effect is lost in the final challenge session (Fig. 5d). Interestingly, P45 Hx mice continue to display altered stepping patterns (Fig. 5c, d), despite showing improved adaptive learning (Fig. 4). Thus, neonatal injury results in age-related changes not just in adaptive learning, but also in stepping patterns.

As PCs are critical to adaptive conditioned learning responses, our in vivo studies targeted at PCs address mechanisms more closely associated with the neurobehavioral readout obtained using the Erasmus ladder. Here, we have analyzed both simple spikes and complex spikes. Simple spike discharge is caused by intrinsic PC activity[58] as well as by parallel fiber (PF) stimulation of PCs via mossy-fiber-granule cell connections[59]. Complex spikes are caused by CF-to-PC circuit activity[60]. Thus, as the cerebellar cortical circuitry matures, simple spike and complex spike frequencies are also expected to change[61]. Our observation of an increase in basal and evoked simple spike firing frequency between P13 and P21 Nx animals (Fig. 6 c–e) is comparable with previous in vivo analyses of PC firing[41,61]. Importantly, our observation that basal and evoked PC simple spike firing frequency is drastically reduced in P13, P21 (Fig. 6 e), P30 (Fig. 8 c), and even in P45 Hx groups (Supplementary fig. 8 b) is significant, because changes in PC firing patterns are an important feature of different mouse models of ataxia[42,62,63]. Although both simple and complex spike firing may be affected in ataxia and other cerebellar disorders, there is evidence to indicate that there are differential alterations in spike discharge patterns in multiple models, which may contribute to the manifestation of behavioral deficits[64,65]. Data from these studies indicate potential mechanisms for the reduction in PC simple spike firing observed in the Hx model, involving either alterations in synaptic input (extrinsic) or changes intrinsic to PCs. In our data, we found changes only in evoked complex spike firing patterns in Hx animals (Fig. 9), whereas basal complex spike firing frequency remains unchanged. As both basal and evoked simple spike firing is dramatically affected in our model, it is possible that Hx affects PF-to-PC pathway via mossy-fiber-GC synapses, or spontaneous PC activity owing to cell-intrinsic changes. Our in vitro electrophysiology data suggest that PCs in Hx and Nx groups are largely similar in terms of passive properties (Supplementary table 1). However, the current-frequency curve is significantly different between Hx and Nx (Supplementary fig. 2 e–f) with a lower Y0 in Hx PCs (0.92 Hz) than Nx PCs (2.32 Hz), supporting the contention that differences in basal spiking maybe cell intrinsic.

In our previous study, we showed that developmental disruption of the [Cl⁻]-accumulating co-transporter, NKCC1, mimics Hx[29]. Interestingly, a recent study showed that abnormal simple spike PC firing owing to disrupted [Cl⁻] homeostasis may be responsible for an increase in short step percentage in adult mutant mice on the Erasmus Ladder[66]. This is consistent with our behavioral data from P45 Hx mice, showing higher number of short steps compared with Nx P45 mice (Fig. 5). However, in contrast to the previously referenced study, we found reduced simple spike activity in Hx animals in vivo, suggesting that [Cl⁻]-accumulating rate acts as an analog-dial modulating the rate of simple spike generation. Importantly, this indicates that factors contributing to [Cl⁻] gradient maintenance, such as early GABA signaling, can be therapeutically targeted in developmental injury.

Previous studies of therapeutic interventions in preclinical rodent models of neurodevelopmental locomotor disorders have typically used tests such as the rotarod task[67], balance beam task[68], inclined beam task[69], or ink/dye footprint assay[70]. We have used the Erasmus Ladder as a system to analyze therapeutic intervention of targeting the developing GABA system to ameliorate locomotor and cerebellar learning deficits. Our time window of intervention (P12–P16) spans the lag end of the early developmental depolarizing GABA phase moving into the more mature inhibitory GABA phase[15]. That treatment with the GABA reuptake inhibitor Tiagabine only rescues locomotor performance by reducing missteps (Fig. 7a)—but does not affect adaptive cerebellar learning (Supplementary fig. 7)—suggests that increasing GABA immediately following early injury affects circuits or cells more generally related to locomotor performance, possibly linked to WM improvement[71].

Consistent with previous findings, a key factor potentially affecting adaptive learning in Hx mice is the reduced basal firing rate of PCs. However, our optogenetics data also indicate reduced evoked firing rate of PCs at all ages studied (Fig. 6c–e, Fig. 8b, c). Whereas spontaneous PC activity is critical for the formation of cerebellar circuitry in postnatal development[15] and locomotor performance[42], within the context of circuit mechanisms responsible for cerebellar learning, the mossy-fiber-to-PC via PFs, and the inferior-olive-to-PC via CFs are also important, as these circuits encode the CS and US, respectively[64,72], in associative cerebellar learning. Emerging data suggest that in addition to the CF and PF cerebellar systems, evoking PC simple spike activity per se can also drive some forms of motor learning such as the VOR[44]. However, given that MLIs are necessary for normal acquisition of adaptive cerebellar learning via synaptic inhibition of PCs[73,74], we cannot completely rule out the potential contribution of MLIs to the abnormal firing properties we observe.

Our electrophysiological data indicate that Tiagabine increases basal firing rates of PCs, which may reflect an improvement in pacemaking activity involving P/Q type $Ca^{2+}$ channels[42] and consequently improve motor performance. However, Tiagabine treatment does not markedly improve evoked PC simple spike firing, which may explain why learning does not improve (Supplementary fig. 7) in Tiagabine-treated Hx animals. In addition, our data showing that adaptive learning does not significantly improve in Tiagabine-treated animals (Supplementary fig. 7), combined with the recovery of evoked complex spike firing patterns in treated animals to near-normal levels (Fig. 9 f), further support the idea that faulty simple spike generation in PCs contributes to long-term adaptive cerebellar deficits, as observed using the Erasmus Ladder. Although we cannot negate the contribution of MLIs, It is tempting to speculate that adaptive learning and locomotor coordination driven by PC activity per se is disrupted owing to neonatal brain injury. Further experiments are needed to address this important question. Although there may be other mechanisms by which Tiagabine partially recovers PC activity in the context of neonatal brain injury, our findings that the GABA reuptake inhibitor ameliorates both the

physiological and the behavioral phenotypes caused by Hx point to the GABAergic network as a potential therapeutic target for prematurity-related locomotor deficits.

## Methods

**Animals**. Female and male wildtype C57BL/6 mice were used for behavior experiments. Female and male *Pcp2-cre* mice (B6.129-Tg(Pcp2-cre)2Mpin/J; The Jackson Laboratory, Bar Harbor, ME, USA) were used for in vivo optogenetics and multielectrode array recording experiments. All animals were handled in accordance to the Institutional Animal Care and Use Committee (IACUC) of Children's National Medical Center, who approved all protocols, and the Guide for the Care and Use of Laboratory Animals (National Institutes of Health).

**Hypoxic rearing**. To simulate the effect of neonatal brain injury in a mouse model, we used chronic sub-lethal neonatal hypoxic rearing. In brief, P2 C57BL/6 mouse pups were cross-fostered with CD-1 mothers and placed in a hypoxic chamber (10.5% $fiO_2$) from P3–P11 (unless otherwise mentioned). After Hx, mice were returned to standard laboratory housing conditions following a brief period of reacclimatization to Nx conditions.

**Behavioral testing**. Cerebellar behavior was studied at two ages—P25 and P45 (age on first day of testing). All mice used for behavioral experiments were naïve and had not been subjected to prior behavioral testing.

**Erasmus Ladder test parameters**. All behavioral tests to measure cerebellar function was performed on the Erasmus Ladder (ERLA-0010; Noldus Information Technology bv, Waginengen, The Netherlands). Erasmus Ladder software (v1.0 and 1.1) was used to control the Ladder. We used the default ladder protocol (Fig. 1a) for paradigms 1 and 3 to test cerebellar behavior at P25 and P45 respectively. Experimental protocol for paradigms 1 and 3 consisted of four training sessions followed by four challenge sessions, with one session performed per day. Each training session (sessions 1–4; Fig. 1c) consisted of 42 "unperturbed" trials where no obstacle challenge was presented. During challenge sessions (sessions 5–8; Fig. 1c), the 42 trials were presented as randomly sequenced categories of "unconditioned stimulus only" (US-only), "conditioning stimulus only" (CS-only), or "paired". For the US-only trials, animals confronted a computer-controlled obstacle (US; Fig. 1b)—which consisted of a randomly activated obstacle to block the path of movement at some point along the ladder. For CS-only trials, a high-pitch warning tone was randomly presented while the animal was at some point along the ladder. And for paired trials, CS was presented followed by US with an interstimulus interval of 250 milliseconds (Fig. 1 b, c). For paradigm 2, we used an extended version of the standard protocol where instead of four challenge sessions, we increased the number of challenge sessions to eight.

**Optogenetic targeting and in vivo electrophysiology**. In order to study firing properties of PCs, we used in vivo optogenetics coupled with multielectrode array recording. For all optogenetics experiments, we employed a *Cre*-dependent channelrhodopsin-mCherry adeno-associated virus (AAV) (Supplementary figure 1a) (AAV9.EF1.dflox.hChR2[H134R]-mCherry.WPRE.hGH; UPenn Vector core, Cat # AV-9-20297P), originally developed by the Karl Deisseroth Laboratory at Stanford University. For in vivo extracellular recordings, we used 1 MΩ tungsten electrode arrays of eight electrodes (Microprobes MEA, MD USA) coupled with a 200 μm diameter optical fiber connected to a 473 nm wavelength LED light (PlexBright LED 4 channel optogenetic controller with Radiant software, version 2.0, Plexon, USA). The electrode array was stereotactically (A/P −5.2 mm; M/L −2.1 mm; D/V −0.35 mm) inserted into the cerebellum of isoflurane anesthetized, head fixed, virus-injected *Pcp2-cre* mice to selectively stimulate PCs, whereas simultaneously recording their spontaneous and optically stimulated evoked spike activity. The same electrode position was used in all electrophysiological experiments.

**Drug injections**. The GABAergic reuptake inhibitor, Tiagabine (Cat. No. 4256, Tocris, Bristol, United Kingdom) diluted in 0.5 × saline solution, was administered to the pups via i.p injection from P12–P16 at a dosage of 50 μg per g body weight. A single injection of the drug solution warmed to 37 °C was administered per day at an injection volume of 10 μl per g body weight.

**Data presentation and statistical analysis**. Prism 6 (GraphPad) and OriginPro 2015 (OriginLab) was used to analyze data obtained from Erasmus ladder experiments. Statistical analysis was conducted using Prism 6. For comparing missteps and backsteps across groups, Two-way analysis of variance (ANOVA) with Sidak's multiple comparison was performed. For comparing steptimes within and across groups, Two-way repeated measures (RM) ANOVA with Tukey's multiple comparison was performed, and *P* values were reported. To obtain normalized learning, post-perturbation steptimes recorded for an individual mouse were normalized to the post-perturbation steptime recorded for the same mouse during session 4. The same statistical test used for absolute

learning was employed for comparing normalized learning. For comparing step classifications across age, we used Two-way ANOVA with Tukey's multiple comparison. For all behavioral experiments, two criteria were used to determine outliers rejected from group analysis: post-perturbation steptime measurement for session 5, and/or percentage of jumps in the last four sessions. ROUT method was used to determine outliers for both criteria ($Q = 1\%$). For optogenetics and in vivo electrophysiology data quantification and visualization, Origin 2016 (Origin lab) was used. For electrophysiology data analysis, Prism 6 (GraphPad) was used. Firing frequency was analyzed in a time-window of 1000 ms prior to optostimulation (pre-optostimulation) and 1000 ms after the start of optostimulation (post-optostimulation). Pre-optostimulation and post-optostimulation frequency sampling consisted of 10 and 11 successive time-points, respectively. One-way ANOVA with Tukey's multiple comparison test was used for statistical hypothesis testing, and multiplicity adjusted *P* values were reported. For comparison of exponential fitting parameters in current-frequency plot, Extra sum-of-squares *F* test was used to compare Y0 and k in the equation $Y = Y0e^{(kX)}$. For statistical analysis of mCherry$^+$ cell counts, two sample or two-tailed *t* test was used. For comparing non-PC:PC cell count ratio, Fisher's exact test was used. $\alpha = 0.05$ was set for all statistical hypothesis testing. Adjusted *P* values were reported for all multiple comparison tests. Sample sizes were not pre-determined based on a statistical test, but are based on standard ranges set in the field. For behavioral tests, a small minority of measurements in some sessions (within the same experiment) did not follow a Gaussian distribution as estimated by normality testing, however, we used RM-ANOVA or two-way ANOVA for all behavioral experiments for purposes of consistency, as the data represented successive measurements from the same sample. Animals from at least two litters per group were used for all experiments. Blinding or randomization was not possible. As behavioral experiments were fully automated, animal-experimenter interaction bias was potentially negligible[75].

**Data availability**. Data included in this manuscript will be made available by the corresponding author based on reasonable request.

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

## Acknowledgements

We especially thank Reinko Roelofs (Noldus Inc.) for technical help and support for the Erasmus Ladder. We thank Dr. Joseph Scafidi and Dr. Li-Jin Chew for critically reading the manuscript. We also thank Dr. Zenaide Quezado (former Director), Dr. Joshua Corbin (current Director), and Dr. Li Wang for leadership and management of the Animal Neurobehavior Core at Children's Research Institute. This work was supported by IDDRC U54HD090257 (V.G.).

## Author contributions

A.S. and V.G. conceptualized the study. A.S., S.K., and V.G. designed experiments. A.S. performed all behavioral experiments and drug injections. S.K. performed all in vivo optogenetics electrophysiological recordings. J.A. performed all in vitro electrophysiological recordings. A.S., S.K., and J.A. performed data analysis. A.S., S.K., and V.G. prepared the figures. A.S. and V.G. drafted the original manuscript, with contribution from S.K. A.S. and V.G. critically revised the manuscript for intellectual content. V.G. coordinated the entire project.

## Additional information

**Competing interests:** The authors declare no competing interests.

