## [Peer Review File · Nature Communications]

Reviewers' comments:

Reviewer #1 (Remarks to the Author):

MANUSCRIPT: NCOMMS-17-16881

TITLE: Neonatal brain injury causes cerebellar learning deficits and Purkinje cell dysfunction

AUTHORS: Aaron Sathyanesan, Srikanya Kundu, and Vittorio Gallo

In this manuscript, the authors used a combination of behavior, in vivo electrophysiology, and optogenetics to examine motor circuit dysfunction in a mouse model of neonatal hypoxic injury. Using the Erasmus Ladder they show that motor learning is deficient and with electrophysiology conducted in postnatal and adult mice, they show that cerebellar Purkinje cell activity is disrupted. With an optogenetics paradigm they further show that evoked activity is also compromised. Finally, using in vivo pharmacology they demonstrate that GABA signaling deficits mediate the Purkinje cell defects. This is a very interesting manuscript conducted by a group of highly talented researchers. The question is important and the model, neonatal hypoxia, is one created by the investigators. Overall, I feel that the authors present a unique model system and perform a beautiful series of experiments. These data could greatly add to the cerebellum literature and there is no question that this study unveils new directions for hypoxia research.

Stylistically, the writing of the body of the manuscript is mostly clear (see below for specific concerns), although I should say that it is very accessible to a wide audience. The majority of the figure panels are easy to follow (again see below for specific comments).

I have a number of comments that will help improve the clarity of the paper.

Major Concerns:

1) The discussion needs a major overhaul. First of all, 8.5 pages of discussion is much too long and unnecessary. Second, the discussion jumps around from topic to topic with no clear take home message. This section needs to be completely restructured. Third, these two points above are reflected in the fact that there are 137 references – I think many of these can be cut.

2) Although the Erasmus Ladder data are appealing, it is presented in an overly complicated manner. I suggest that the authors rewrite these sections in order to make the main message of each section clear and concise. As it stands, there are so many data points that were seemingly conflicting in the young versus the old, or just simply minimal statistical power. The authors then sum each section with statements that support differences along side statements that say no difference was found. It is just very confusing to get past this part of the manuscript. Additional comments on this are provided below.

Additional comments:

1) The term "metamorphosis" is used at least twice in the paper to refer to how the

cerebellum develops. This is a strange term and perhaps best left for insect development.

2) On page 4, why are the words adult and developing in italics? I see no reason for this.

3) On page 4 of the introduction the authors state "GABA plays distinct roles including excitatory neurotransmission..." This has not been demonstrated in cerebellum.

4) In the first section of the Results the authors state "we tested for locomotor coordination deficits in mice at two weeks following hypoxic rearing." But, at what exact age was this analysis conducted?

5) The authors conclude the first section of the Results by stating "Thus, neonatal brain injury causes an initial locomotor miscoordination in mice, as indicated by a higher baseline percentage of missteps compared to controls." It is not clear why there is only an "initial" defect. Are the authors suggesting that some kind of compensation occurs?

6) On page 9 the authors state "We predicted that lengthening the challenge session would increase the likelihood of seamless associative learning, eventually resulting in little difference between pre-perturbation and post-perturbation steptimes within the Hx group." I have no idea what you mean by this. Also, I do not understand what lead you to predict this in the first place.

7) On page 12 the authors conclude "Thus, locomotor stepping patterns in younger Hx animals displayed a propensity towards normal behavior over training, however this gradually reverted by the end of challenge sessions. In older Hx animals, stepping patterns during training sessions stayed relatively the same, changing slightly during the challenge sessions, but overall still significantly different from the Nx group." Again, I have no idea what you are concluding or for that matter based on the data I cannot understand why this is the case.

8) On page 12 the authors state "Previously, we had observed abnormal synaptic activity of GABAergic interneurons in cerebellar white matter of mice following Hx." I don't quite follow, are GABAergic interneurons ectopically located in the white matter? Which ones? How does this happen? What was the timing of the manipulation to get this result?

9) The authors have used evoked activity driven by optogenetics to examine Purkinje cell function. Although this is a clever idea, I don't think I fully appreciate the physiological relevance of this. What is the evoked paradigm telling you over the spontaneous activity? Why not just record in awake animals instead? I don't understand how you would link this data to the behavior deficits you see. To some degree it certainly mimics what people do in slice where axons are severed by the preparation, etc, but by ramping up activity with such a powerful paradigm like ChR2, what can you actually conclude about the state of those neurons? The authors need to provide much more background rationale for this approach and some sense of what you are doing to these neurons would be helpful. For example, with the light on, do you only get activation? Did you need see any depolarization block of Purkinje cells? In sum, I am just a little weary about using optogenetics to ask cellular level

questions. It is great as a circuit mapping tool, but when you combine it with cell specific behaviors plus the complicated activity of a developing neuron, the results become much more difficult to parse out.

10) Along the same lines as above, the authors state "Optostimulation was presented for a total of 1000 ms." Why? How was this value chosen? What happens at other durations?

11) What was the intensity of blue light delivered in each case? Along the same lines as this, on page 15 the authors state "We then developed an optimized optostimulation paradigm...". This paradigm needs to be fully explained here. What was actually optimized?

12) On page 13, the authors state "In comparison, following Hx, PCs at P13 (Figure 6e, red open circles) as well as P21 (Figure 6e, wine open circles) displayed severely reduced basal firing frequency compared to PCs in age-matched Nx controls (Tukey's multiple comparison test following One-way ANOVA, P13 Nx basal vs. P13 Hx basal: $P < 0.0001$; P21 Nx basal vs. P21Hx basal: $P < 0.0001$)." Okay, but what were the actual rates of firing in the Hx cells?

13) In regards to the data in point number 12 above, was the analysis conducted for simple spikes plus complex spikes combined? If yes, why?

14) The authors conduct several analyses for Purkinje cell "spike profiles". It is not clear at all what this analysis provides over the accompanying analysis of frequency, CV, and CV2. What problem are you actually looking to solve with this measurement?

15) On page 15, the set up or rationale for optogenetic evoked measurements is stated as "Evoked PC firing has been implicated in the temporal aspects of associative motor learning^{49, 50}." Similar to what I have already mentioned, I do not understand how the otogenetic evoked responses can be related to behavior. The References that are cited here for eye blink conditioning are based on ferrets, and other species. So, how do we compare your developmental data in mice using a tool that drives VERY high levels of activity, to an entirely different method, for a different behavior, conducted in a different species, and at a different age?

16) On page 16 the authors state "Finally, we also confirmed that the electrode position was at the PC soma layer..." What lobules were you analyzing? Was this consistent from animal to animal? That is, was a specific lobule (or lobules) targeted and then consistently used for the analysis?

17) On page 20, the authors state "Surprisingly, however, in contrast to our observations of simple spike firing between groups, we did not observe a significant difference in mean basal complex spike firing frequency between Nx and Hx untreated groups..." Please explain why this is surprising. In any case, I am not sure what you are trying to conclude from this data.

18) The quality of the multi-electrode array recordings is not clearly show. That is, on each

session for each array, on how many channels do you successfully isolate cells? What do these look like? Simultaneously recorded sample traces should be shown. This will help justify why this approach was taken to begin with (ie. Why not just use single electrodes?).

19) From the raw data that is shown, it is not so convincing whether complex spikes and simple spikes can be reliably isolated. The authors need to demonstrate the quality of traces in full (see above) and also show overlays of several dozen spikes from individual traces to show the consistency in waveform per recording.

20) The raw traces in Fig. 6 and 8 are cut off at the top and sometimes at the bottom, too. Please go back to your original files and replace the data to show the spikes from baseline to true peak.

21) In Fig 6 panel B, Nx raw trace, how come the firing rate is still elevated after the light is off? Perhaps this is not an ideal example, but that brings me to my next point, the authors need to somehow show using raw traces and additional graphs the consistency in the responses of the cells to ChR2. The line graph shown for cumulative response is convincing, but this needs to be backed up by additional data for some individual cells. We need to get a better appreciation of the actual data set.

22) Fig. 6 Hx cell, how do you prove that you were actually stimulating this cell yet it was truly unresponsive? What if you were just simply too far away with the light source, or the cell was not properly isolated? Where are the complex spikes in the Hx trace – based on what is shown I cannot see any?

23) Fig. 9B, the period of increased CS activity should be show on a raw trace. Therefore, the authors need to provide a period of recording that is long enough to show the abrupt transition from light off to light on.

24) Fig. 9A, we need to see the adjacent SS activity as well. So, the authors also need to provide a slightly lower power trace to go along with what is already provided. Just the CS's blown up on their own gives me very little information about what else the cell is doing in this condition.

25) Fig. S3, the merged images on the right of panels a and b are much too small. Also, please label the lobule(s) that is shown in panels c and d.

Reviewer #2 (Remarks to the Author):

The manuscript by Sathyanesan and colleagues describes the long-term consequences of neonatal hypoxia for the young adult organism. For that purpose, they took advantage of an experimental mouse model in which, for a few days, perinatal mice are reared under hypoxic conditions (lowering oxygen concentration to 10 % rather than 20 %). This work is an extension of their previous study (Zonouzi et al., 2015).

Here, the authors focused on the analysis of behavioral alterations using automated assessment of motor behavior by the Erasmus ladder. The mice have to run along two rows of alternately positioned rungs. The automatic control senses not only the weight on individual rungs while the mouse is running along the ladder, but can also insert obstacles. Thereby, this device cannot only quantify impairment of motor performance, but also of associative and adaptive motor learning. These experiments were complemented by optogenetic and pharmacological experiments.

The main findings are:

1. Perinatal hypoxia induces a significant impairment of motor performance and of cerebellar learning.
2. In young adult mice, the behavior can partially be rescued by (1) evoked neuronal photo-stimulation or (2) inhibition of GABA reuptake.

While the first result might be expected, the second result provides a strong translational aspect. Modulation of GABAergic signaling might be a potent target for future drug development and treatment of pre-term infants.

The experiments are well performed, and the results are important. The manuscript itself is concisely written. The discussion section is very stimulating for future research.

Although this work adds to a previous study, there are still open questions: Zonouzi et al. demonstrated impaired NG2 cell proliferation and differentiation, while here the physiology of Purkinje cells was studied. The authors left it to future studies to provide a solid experimental link between impaired NG2 and Purkinje cell activity.

Minor points:

Although the statistical values are given, a supplementary table listing the precise numbers of mice tested per individual experiment would be very helpful to estimate the statistical power. The authors do not provide scatter plots for all figures (e.g., they are missing in figure 2-4, and 7, but are depicted in figure 5 or for the optogenetic and pharmacological data sets).

Reviewer #3 (Remarks to the Author):

The authors have performed an in depth analysis of the behavioral consequences of cerebellar alterations resulting from hypoxic brain injury in the newborn, using a murine chronic hypoxia model.

They have used an Erasmus Ladder test to study cerebellar behavior and demonstrate that hypoxia leads to locomotor malperformance and also cerebellar learning deficits. They use an optogenetic approach for the first time in a chronic hypoxia model system to show that hypoxia leads to Purkinje cell firing frequency dysfunction. Moreover, they demonstrate that the GABA-reuptake inhibitor Tiagabine partially restores the defective locomotor performance and improves PC firing.

This is a technically very good and thorough paper, introducing several new techniques, both behavioral and pathophysiological, for evaluating cerebellar dysfunction in the context of hypoxic injury.

I would recommend a more detailed discussion of other non-cerebellar brain regions which might also be affected in hypoxia, and contribute to the locomotor dysfunction.

Reviewer #4 (Remarks to the Author):

In this report, Sathyanesan and colleagues find that neonatal hypoxia exposure results in motor dysfunction and motor learning abnormalities that point to cerebellar injury. In vivo characterization of Purkinje cell (PC) activity revealed simple spiking deficits in hypoxic (Hx) mice. Behavioral and functional deficits could be partially reversed by pharmacological perturbation of the "developing GABA system". The causal mechanisms underlying pharmacological amelioration of PC spiking and motor behavior deficits remained unexplored. Provided the authors' previously reported observations of Hx-induced connectivity changes in the cerebellum and accompanying changes in PC intrinsic excitability (measured in vitro) (Zonouzi et al. 2016), the findings in this report are rather unsurprising. Perturbation of PC firing is well known to affect movement and motor learning. That multi-day treatment with a GABA uptake blocker (Tiagabine) can partially ameliorate the effects of neonatal Hx are, again, not surprising as the authors have already reported that such treatment reverses cellular-level effects of hypoxia on GABAergic circuitry in the cerebellum (Zonouzi et al. 2016). By missing an opportunity to provide further insight into how GABA uptake block leads to partial restoration of cerebellar function and its influence on behavior, the authors fail to provide useful information that may lead to the further development of therapeutic interventions for Hx-induced brain injury. Overall, this work is unlikely to find a wide audience outside of the hypoxia field. Also, the work suffers from several potential issues with experimental design.

1. Motor deficits are presumably due to PC spiking changes DURING motor behavior. However, the effects of Hx are only explored in vivo on intrinsic excitability under anesthesia (how is this measure really different than that that already performed in slices by the same group?). This is a major limit to the conclusions drawn in the present work especially if motor performance and learning deficits are to be related to PC spiking abnormalities.

2. Regarding measurements of ChR2-induced spiking in PCs, the authors face a potential confound in that they have not adequately controlled for Hx-induced changes in ChR2 expression (perhaps neonatal hypoxia exposure diminishes ChR2 expression levels in PCs compared to Nx controls). Although the authors attempt to control for this possibility by looking at mCherry fluorescence, this is an indirect method. A more well-controlled experiment would be to measure evoked photocurrent amplitudes in Hx and Nx treated mice.

3. Complex spikes are evoked by the activity of climbing fibers. So why does optogenetic stimulation of PCs induce complex spiking? An indirect effect through disinhibition of the inferior olive would be expected to be accompanied by a delay- this is not apparent (or quantified) in their data.
4. If diminished PC simple spike output is a key variable explaining Hx-induced locomotion and motor learning deficits, can this be recovered by optogenetic stimulation of PCs during behavior?
5. Is the Hx effect on PC excitability ubiquitous across the cerebellum? PCs in zebrin-delimited zones show extensive functional differences including in their intrinsic excitability. This should be assessed. Relatedly, are the cerebellar areas that were recorded from even related to the motor behaviors studied in their task?
6. The Pcp2-Cre line used in the study should be identified (Jackson Lab maintains two different lines; Jdhu and Mpin). Notably, Cre activity in the Mpin line has been shown to be non-specific for PCs (see Witter et al. and Regehr 2016 [Neuron] who used functional assessments with ChR2-induced excitation which is more sensitive than an XFP reporter as used in the present study; also see Zhang et al. 2004 [Genesis]). If the authors used the Mpin line, the lack of PC specificity could influence the outcome of their results.
7. Notably, the authors observe recovery over time for locomotor deficits (i.e. at P45). Are compensatory mechanisms for this recovery observable at the circuit level? If so, do they match that induced by Tiagabine (e.g., are PC simple spiking deficits restored)?
8. Is the effect of Tiagabine on ameliorating motor deficits specific for the cerebellum (the authors use IP injection of Tiagabine which will have a global influence)? Why not apply Tiagabine locally (e.g., by cannulae) to bolster the conclusion that recovery is specific for this brain area?
9. How is Tiagabine working to partially restore PC spiking? How does this relate to "targeting the developmental GABA system"? Is there an increase or decrease in GABA levels with this pharmacological manipulation? Is GABA uptake block affecting cortical GABAergic circuitry (OPCs, Golgi cells, stellate/basket cells)? Purkinje cells themselves are also GABAergic so GABA re-uptake block in the deep cerebellar nuclei might also effect their influence on behavior.
10. Presumptively, experience-dependent improvement in misstep performance for both Nx and Hx mice during the initial training phase (session 1-4) reflects some type brain-dependent function. Is the locus known?

Response to Reviewers' Comments on "Neonatal injury causes cerebellar learning deficits and Purkinje cell dysfunction"

Aaron Sathyanesan, Srikanya Kundu, Joseph Abbah, and Vittorio Gallo

We thank the reviewers for their valuable and detailed comments on our manuscript entitled "*Neonatal brain injury causes cerebellar learning deficits and Purkinje cell dysfunction*". As the reviewers will notice, we have substantially revised our manuscript, incorporating changes suggested by the reviewers, and providing further clarification either in the manuscript itself or in this response. Our revised manuscript thus contains changes to certain sections of the text, as well as additional data and figures where appropriate. Appropriate changes have been highlighted in the manuscript. Below, we provide a point-by-point response to our reviewers. Comments by reviewers are italicized and are marked as 'C' followed by a number and our response to a particular comment are marked as 'R' followed by the same number:

Reviewer #1:

Summary

C1. *"Overall, I feel that the authors present a unique model system and perform a beautiful series of experiments. These data could greatly add to the cerebellum literature and there is no question that this study unveils new directions for hypoxia research."*

R1. We thank the reviewer for their enthusiastic and positive overall assessment of our manuscript.

Major concerns

C2.1. *The discussion needs a major overhaul. First of all, 8.5 pages of discussion is much too long and unnecessary. Second, the discussion jumps around from topic to topic with no clear take*

home message. This section needs to be completely restructured. Third, these two points above are reflected in the fact that there are 137 references – I think many of these can be cut.

R2.1. We appreciate the reviewer's candid assessment of our discussion section. While we accept that our discussion section is indeed longer than what may be considered "typical", we have received positive assessment of the discussion section from Reviewer # 2. Additionally, Reviewer # 3 would like further treatment on certain aspects of the discussion. To respectfully address these somewhat competing assessments, we have revised our discussion. First, we have shortened the discussion section by 20%. Second, we have highlighted the key take-away messages from each paragraph, while not diluting the overall central theme of the manuscript. Lastly, we have introduced a new paragraph (pp 22-23), incorporating a part from a section in the unrevised manuscript, and further discussing specific potential contributions of non-cerebellar regions to the locomotor deficits we observe in Hx mice.

C2.2. Although the Erasmus Ladder data are appealing, it is presented in an overly complicated manner. I suggest that the authors rewrite these sections in order to make the main message of each section clear and concise. As it stands, there are so many data points that were seemingly conflicting in the young versus the old, or just simply minimal statistical power. The authors then sum each section with statements that support differences along side statements that say no difference was found. It is just very confusing to get past this part of the manuscript. Additional comments on this are provided below.

R2.2. Upon revisiting our Erasmus Ladder results section, with this comment by the reviewer in mind, we agree that the results along-side the detailed statistics may indeed come across as confusing to the reader. We have therefore modified the section for the Erasmus Ladder results

to allow for only key statistical differences to be mentioned in the text. A complete list of the statistical tests and results are provided in the revised figure legends.

Additional comments

C3.1. The term “metamorphosis” is used at least twice in the paper to refer to how the cerebellum develops. This is a strange term and perhaps best left for insect development.

R3.1. The context in which we referred to “metamorphosis” was a direct reference from an excellent review by van Welie et al. (2011) on the developmental changes in the cerebellar cortex at the level of cellular physiology and circuitry¹. However, we agree that to a wider audience, the term may come across as misplaced. Hence, we have changed “metamorphosis” to “maturation”:

PC circuit maturation engenders normal cerebellar behavior at later stages. Less understood, however, are specific features of PC function and resultant cerebellar behavior which might be disrupted due to early hypoxic insult during circuit maturation.

C3.2. On page 4, why are the words adult and developing in italics? I see no reason for this.

R3.2. We wished to highlight the potentially different role GABA plays in the developing cerebellum, where GABA is excitatory and may act as a developmental signal, in contrast to the adult cerebellum, where GABA is an inhibitory neurotransmitter. However, we accept this criticism and have removed the italicization.

C3.3. On page 4 of the introduction the authors state “GABA plays distinct roles including excitatory neurotransmission...” This has not been demonstrated in cerebellum.

R3.3. We respectfully disagree with the reviewer, since GABA-induced neuronal excitability in the developing cerebellum has been demonstrated in Purkinje cells^{2,3}, molecular layer interneurons⁴, as well as granule cells^{5,6}. To further clarify the role of GABA in the developing cerebellum, we have replaced “excitatory neurotransmission” with “neuronal excitation” since this captures the effect of depolarizing GABA in synaptic as well as extrasynaptic contexts. Relevant citations pertaining to Purkinje cells have been added to the text.

C3.4. In the first section of the Results the authors state “we tested for locomotor coordination deficits in mice at two weeks following hypoxic rearing.” But, at what exact age was this analysis conducted?

R3.4. This analysis was conducted at P25. This has been clarified in the text.

C3.5. The authors conclude the first section of the Results by stating “Thus, neonatal brain injury causes an initial locomotor miscoordination in mice, as indicated by a higher baseline percentage of missteps compared to controls.” It is not clear why there is only an “initial” defect. Are the authors suggesting that some kind of compensation occurs?

R3.5. What we meant by “initial” defect referred to the first two sessions of the Erasmus Ladder experiment where missteps were measured (Figure 2A; sessions 1 and 2 – ‘training’). Although we cannot exclude the possibility that this may be a form of short term compensation, this is not what we were suggesting. Rather, we wished to highlight that Hx mice, on average, are able to perform as well as Nx mice from sessions 3-8 in terms of percentage of missteps. And so, our results in terms of the cerebellar learning paradigm, where we do see differences between Hx and Nx mice at P25 are not due to a general lack of locomotor coordination.

C3.6. On page 9 the authors state “We predicted that lengthening the challenge session would increase the likelihood of seamless associative learning, eventually resulting in little difference between pre-perturbation and post-perturbation steptimes within the Hx group.” I have no idea what you mean by this. Also, I do not understand what lead you to predict this in the first place.

R3.6. The challenge sessions denote the sessions wherein we present the mice with paired US-CS trials to test associative cerebellar learning. Cerebellar-dependent eyeblink conditioning experiments have shown that conditioned responses (CRs) in normal wildtype C57BL/6 mice improve over consecutive sessions^{7, 8}. In mice where cerebellar function is altered in some way, cerebellar eyeblink conditioning is delayed, but not completely abolished⁹ suggesting that in some cases, the cerebellar learning curve may at some number of sessions reach the “normal” level of performance. This was the assumption behind our extended challenge sessions. In the case of the Erasmus Ladder, since we see no statistically significant differences between pre-perturbation and post-perturbation steptimes in Nx mice after just one challenge session (day 5), we tested if the difference between pre- and post-perturbation in Hx mice is something that may eventually reduce if the number of challenge sessions are increased.

C3.7. On page 12 the authors conclude “Thus, locomotor stepping patterns in younger Hx animals displayed a propensity towards normal behavior over training, however this gradually reverted by the end of challenge sessions. In older Hx animals, stepping patterns during training sessions stayed relatively the same, changing slightly during the challenge sessions, but overall still significantly different from the Nx group.” Again, I have no idea what you are concluding or for that matter based on the data I cannot understand why this is the case.

R3.7. This conclusion is based on the data presented in Figure 5 where we compare stepping patterns of P25 and P45 mice in both the Hx and Nx groups. What we mean by a “propensity toward normal behavior” indicates the lack of stepping pattern difference (short steps) in P25 Hx animals in session 4 which is the final session of “training”. By the end of the challenge session (session 8), we observe that the Hx animals display short step patterns closer to the first training session (session 1) indicating a deviation from the “normal” performance of Nx animals which is a reduced number of short steps.

However, we understand that our verbiage may be improved. We thus have modified these sentences to:

Thus, locomotor stepping patterns in younger Hx animals displayed a trend towards normal behavior over training, however this gradually reverted by the end of challenge sessions. In older Hx animals, stepping patterns during training sessions stayed relatively the same (~80%). During the challenge sessions, short step percentage in P45 Hx animals reduced to less than 60%, but remained significantly higher than P45 Nx group short step percentage.

C3.8. On page 12 the authors state “Previously, we had observed abnormal synaptic activity of GABAergic interneurons in cerebellar white matter of mice following Hx.” I don’t quite follow, are GABAergic interneurons ectopically located in the white matter? Which ones? How does this happen? What was the timing of the manipulation to get this result?

R3.8. In our previous work¹⁰ we had shown that migrating GAD65⁺ interneurons in the cerebellar white matter exhibit putative functional synaptic contacts with NG2⁺ oligodendrocyte precursor cells (OPCs). The timing of the manipulation (chronic perinatal hypoxia) in our previous paper is the same as described in the methods section of our current manuscript i.e. P3-P11.

C3.9. *The authors have used evoked activity driven by optogenetics to examine Purkinje cell function. Although this is a clever idea, I don't think I fully appreciate the physiological relevance of this. What is the evoked paradigm telling you over the spontaneous activity?*

R3.9. Since its discovery, although optogenetics has been used to great effect as a circuit-mapping tool¹¹, using optogenetics to assess neuronal excitability and firing response dynamics is a relatively new and less explored approach. Since disease states have often been associated with alterations in intrinsic neuronal properties like excitability¹², we have used an optogenetic approach to selectively define intrinsic PC firing properties *in vivo*. The cellular specificity and millisecond precision that can be achieved using tools like ChR2 makes this particular application of optogenetics quite useful. It is worth mentioning that a similar optogenetic approach has been used by another group to investigate neuronal excitability changes in the mouse cortex following global ischemia¹³.

C3.10. *Why not just record in awake animals instead? I don't understand how you would link this data to the behavior deficits you see. To some degree it certainly mimics what people do in slice where axons are severed by the preparation, etc, but by ramping up activity with such a powerful paradigm like ChR2, what can you actually conclude about the state of those neurons? The authors need to provide much more background rationale for this approach and some sense of what you are doing to these neurons would be helpful. For example, with the light on, do you only get activation? Did you need see any depolarization block of Purkinje cells?*

R3.10. The currently design of the commercially available Erasmus Ladder setup that we have used to define behavioral deficits limits our ability to record from animals while they are performing

on the ladder. However, three points must be considered to judge the relevance of the *in vivo* electrophysiology and optogenetic data to our behavioral data:

1. We have performed *in vivo* electrophysiology at the same age and within the same time window as the behavioral experiments (P30 *in vivo* electrophysiology data, Figure 8, and P45 *in vivo* electrophysiology data, Supplementary figure 7). We have also consistently recorded from the simple lobule (Supplementary figure 4 b-c inset), which has been implicated in associative motor learning^{14, 15}. In rats, the simple lobule has also been shown to exhibit developmental increase in eyeblink conditioned responses and simple spike activity¹⁶, which is in line with our observations.

2. Our pharmacological intervention data in P30 and P45 animals shows that photoevoked responses *in vivo* exhibit significant amelioration compared to equivalent responses in P30 and P45 Hx groups respectively (Figure 8 c, Supplementary figure 7 b). Locomotor performance is clearly improved following Tiagabine treatment (Figure 7 a). Thus, the results from our pharmacological intervention acts as a link connecting the behavioral and *in vivo* optogenetics data. It has been proposed that optogenetics approaches may be used in the domain of drug discovery¹⁷. In this regard, our results could be considered an early success in testing the effect of drugs on neuronal hypoexcitability *in vivo*.

3. We acknowledge that using ChR2 to stimulate neurons is indeed a powerful paradigm for stimulation. However, considering that a previous study using ChR2 in PCs *in vivo* reported an upper limit of photoevoked simple spike firing frequency of 250 Hz using 500 ms light pulses prior to depolarization block¹⁸ indicates that the simple spike frequencies which we have observed (34.8 Hz – 76.4 Hz for Nx groups, 4.3 Hz – 49.3 Hz for Hx groups) lie in the lower to middle range of the optostimulation-evoked frequency relationship, especially considering that our light pulse width is only 30 ms. Our optostimulation paradigm is detailed in our methods section.

More to the point of depolarization block: our newly added *in vitro* optogenetics data (Supplementary figure 2 g-j) show that we observed 100% optostimulation fidelity from all PCs we recorded which expressed mCherry-ChR2. In fact, we did not observe depolarization block *in vitro* even for optostimulation lasting a few seconds (Reviewer figure 8). Lastly, we only observed activation or increase in PC firing frequency for all PCs recorded both *in vitro* and *in vivo*.

C3.11. *In sum, I am just a little weary about using optogenetics to ask cellular level questions. It is great as a circuit mapping tool, but when you combine it with cell specific behaviors plus the complicated activity of a developing neuron, the results become much more difficult to parse out.*

R3.11. We acknowledge the reviewer's criticism and caution. However, we would like to direct the reviewer to recent reports wherein optogenetic approaches were used to answer cellular level questions in a developmental context^{19, 20}.

C3.12. *Along the same lines as above, the authors state "Optostimulation was presented for a total of 1000 ms." Why? How was this value chosen? What happens at other durations?*

R3.12. Our spike fidelity calibration plots in supplementary figure 1e and 1f showed that 25 Hz optical stimulation frequency was the optimum for PCs to evoke simple spikes with high (~92%) fidelity. Regarding the duration of the stimulation, 30 ms was the minimum duration needed for those cells to evoked action potentials with high fidelity. Following these two calibrations, we designed our optostimulation paradigm to deliver 25 pulses/sec (25 Hz) with 30 ms duration for each pulse, along with a 10 ms inter-pulse interval. 10 ms inter-pulse interval is sufficient to avoid refractory period saturation between two consecutive evoked action potentials. Based on our

calibration, a much lower ($\ll 30$ ms) or much higher ($\gg 30$ ms) pulse width duration results in a loss of spike fidelity *in vivo* (Supplementary figure 1f).

C3.13. *What was the intensity of blue light delivered in each case? Along the same lines as this, on page 15 the authors state “We then developed an optimized optostimulation paradigm...”. This paradigm needs to be fully explained here. What was actually optimized?*

R3.13. The intensity of blue light was 200mA at the tip of the optic fiber. Regarding the optostimulation paradigm, we would like to direct the reviewer to R3.12 and point #3 in R3.10.

C3.14. *On page 13, the authors state “In comparison, following Hx, PCs at P13 (Figure 6e, red open circles) as well as P21 (Figure 6e, wine open circles) displayed severely reduced basal firing frequency compared to PCs in age-matched Nx controls (Tukey’s multiple comparison test following One-way ANOVA, P13 Nx basal vs. P13 Hx basal: $P < 0.0001$; P21 Nx basal vs. P21Hx basal: $P < 0.0001$).” Okay, but what were the actual rates of firing in the Hx cells?*

R3.14. The basal Hx values for P13 and P21 were 4.3 Hz and 6.85 Hz respectively, which are mentioned in the last sentence of the same paragraph.

C3.15 *In regards to the data in point number 12 above, was the analysis conducted for simple spikes plus complex spikes combined? If yes, why?*

R3.15. No, we performed analysis at P13 and P21 for simple spikes only.

C3.16. The authors conduct several analyses for Purkinje cell “spike profiles”. It is not clear at all what this analysis provides over the accompanying analysis of frequency, CV, and CV2. What problem are you actually looking to solve with this measurement?

R3.16. Alterations in spike shape such as the repolarization/depolarization duration, or slopes are indicative of underlying changes in ionic balance across the membrane, possibly due to ion channel dysfunction or gene expression changes. This in turn affects overall excitability and firing rate of the neuron. For example, the Kv3 subfamily of voltage-gated K⁺ channels enables fast repolarization, thus leading to rapid recovery following an action potential allowing rhythmic high frequency firing²¹. Loss of voltage gated K⁺ channels such as Kcnc3 results in abnormal PC spike shapes, evoked firing frequency, interspike interval, and motor incoordination^{22, 23}. Analogously, with our spike profile analysis we provide evidence that in terms of population-level spike profiles, there are changes in the depolarization and the repolarization states of the PCs after neonatal injury suggestive of alteration in ion channel function, which may underlie changes in firing properties. That the spike characteristics can be partially recovered with pharmacological intervention, along with our observation of a partial amelioration of firing frequency suggests that this recovery is GABA-mediated.

C3.17. On page 15, the set up or rationale for optogenetic evoked measurements is stated as “Evoked PC firing has been implicated in the temporal aspects of associative motor learning^{49, 50}.” Similar to what I have already mentioned, I do not understand how the optogenetic evoked responses can be related to behavior. The References that are cited here for eye blink conditioning are based on ferrets, and other species. So, how do we compare your developmental data in mice using a tool that drives VERY high levels of activity, to an entirely different method, for a different behavior, conducted in a different species, and at a different age?

R3.17. We thank the reviewer for pointing out how our framing of the rationale may have come across as confusing. We have removed the references to non-mouse species and have replaced it with two citations both of which use optogenetics as a means to increase or decrease excitability in PCs and track changes in behavior (locomotor sequences²⁴ and forelimb movement¹⁴). We acknowledge that these cited references still do not track one-to-one with our work regarding the behavioral setup and the age. However, our focus is to assess if neuronal excitability is itself altered due to neonatal injury. Now that we know that PC excitability itself is altered in Hx, from a therapeutic/translational standpoint, this would lead us to design experiments where we may be able to ameliorate excitability (which is what we have done with our Tiagabine treatments), or bypass PCs altogether, in order to rescue behavior in future experiments. In order to clarify our objective, we have modified the framing of our rationale:

PC excitability has been implicated in the temporal aspects of associative motor learning.

Regarding the point about high levels of activity, we would like to direct the reviewer to R3.12 and point #3 in R3.10.

C3.18. On page 16 the authors state “Finally, we also confirmed that the electrode position was at the PC soma layer...” What lobules were you analyzing? Was this consistent from animal to animal? That is, was a specific lobule (or lobules) targeted and then consistently used for the analysis?

R3.18. Yes the stereotactic coordinates for both virus injection and multielectrode array recording are consistent throughout our *in vivo* electrophysiology and optogenetics experiments. We have targeted and recorded from the simple lobule (Supplementary figure 4b-c insets) which has been shown to be associated with motor learning paradigms^{14, 15}.

C3.19. *On page 20, the authors state “Surprisingly, however, in contrast to our observations of simple spike firing between groups, we did not observe a significant difference in mean basal complex spike firing frequency between Nx and Hx untreated groups...” Please explain why this is surprising. In any case, I am not sure what you are trying to conclude from this data.*

R3.19. From the perspective of delay eyeblink conditioning, conditioned responses in PCs are tightly correlated to an increase in complex spike frequency. Therefore, we assumed that complex spike frequency would be significantly altered in Hx animals, given the drastic changes in cerebellar learning as measured by the Erasmus Ladder. The fact that we do not observe major changes in complex spike frequency, suggests that, at least at the level of PCs, the changes in basal simple spike frequency is potentially the major determining factor in the observed behavioral deficits.

C3.20. *The quality of the multi-electrode array recordings is not clearly show. That is, on each session for each array, on how many channels do you successfully isolate cells? What do these look like? Simultaneously recorded sample traces should be shown. This will help justify why this approach was taken to begin with (ie. Why not just use single electrodes?).*

R3.20. To aid our reviewers in better appreciating the multi electrode recordings, representative recording has been shown in Reviewer figure 1. This is an example of multi-electrode recording from a P13 Nx animal. The table in ‘a’ represent the recorded electrodes/channel numbers along with numbers of sorted spikes from each electrode as a unit. In the displayed example, three out of eight channels - channel no. 5, 6 and 8, have successfully recorded isolated cells. Panels b, e and h shows the recorded raw traces, panels c, f, and i show isolated clusters (using PCA) in 2D space for each unit/cell and overlaid waveforms of selected units (marked as colored) over

unsorted (gray) respectively. 'Unit a' in electrode 6 (highlighted as '>>6') shows captured Purkinje cell simple spikes. We have used multielectrode array recordings in order to increase our chances of obtaining usable data from PCs especially using the optogenetic stimulation paradigm.

C3.21. From the raw data that is shown, it is not so convincing whether complex spikes and simple spikes can be reliably isolated. The authors need to demonstrate the quality of traces in full (see above) and also show overlays of several dozen spikes from individual traces to show the consistency in waveform per recording.

R3.21. We have provided a detailed example of spike sorting Reviewer figure 2. Panel 'a' shows a representative raw trace recorded from an Nx P21 animal, which contains both simple spikes, and complex spikes (marked as asterisks). Simple spikes and complex spikes were consistently isolated in Offline Sorter (Plexon V4) by applying different thresholds (as shown as red line in continuous waveforms in panels b and e) along with other isolation criteria for complex spikes mentioned in the methods section. The appearances of the sorted spikes (magenta: simple spikes, blue: complex spikes) within the unsorted spikes (grey) on the continuous recorded time line showed in sorted waveform panel. Combining all of the above isolation criteria, we show a successfully isolated distinct cluster of spikes in 2D space for complex spikes (blue) and simple spikes (magenta) represented in panels 'c' and 'f'. We also show a population of isolated spikes (colored) overlaid on unsorted spikes (grey).

C3.22. The raw traces in Fig. 6 and 8 are cut off at the top and sometimes at the bottom, too. Please go back to your original files and replace the data to show the spikes from baseline to true peak.

R3.22. The raw traces have been replaced both in figures 6 and 8.

C3.23. In Fig 6 panel B, Nx raw trace, how come the firing rate is still elevated after the light is off? Perhaps this is not an ideal example, but that brings me to my next point, the authors need to somehow show using raw traces and additional graphs the consistency in the responses of the cells to ChR2. The line graph shown for cumulative response is convincing, but this needs to be backed up by additional data for some individual cells. We need to get a better appreciation of the actual data set.

R3.23. PC responses to the optostimulation were heterogeneous in terms of their response duration (tailing effect), change in frequencies and decay time. That is why we preferred to show a population response with standard errors instead of an example graph containing few cells. In our hands, the response to optostimulation for a population of 25 PCs indicated a tailing effect.

C3.24. Fig. 6 Hx cell, how do you prove that you were actually stimulating this cell yet it was truly unresponsive? What if you were just simply too far away with the light source, or the cell was not properly isolated? Where are the complex spikes in the Hx trace – based on what is shown I cannot see any?

R3.24. We have conducted *in vitro* slice experiments where we verify that Hx PCs expressing mCherry-ChR2 are viable (Supplementary figure 2 a-f) and respond to optostimulation under patch-clamp mode in a manner similar to Nx PCs with ChR2 (Supplementary figure 2 g-j). It is highly unlikely that the cell was far away from the fiber since we survey multiple sites within the simple lobule for opto-evoked responses and perform post-hoc validation of the electrode position to ensure we are in the right area in terms of mCherry-ChR2 expression. The fact that we see

reduced photoevoked response in Hx PCs *in vivo*, but see normal photoevoked responses *in vitro* suggests that there is a combinatorial effect of the *in vivo* environment and reduced intrinsic neuronal excitability that results in reduced photoevoked responses *in vivo*.

C3.25. *Fig. 9B, the period of increased CS activity should be show on a raw trace. Therefore, the authors need to provide a period of recording that is long enough to show the abrupt transition from light off to light on.*

R3.25. We have provided raw traces of complex spikes as requested by the reviewer in Figure 9 panels 'a' and 'b'.

C3.26. *Fig. 9A, we need to see the adjacent SS activity as well. So, the authors also need to provide a slightly lower power trace to go along with what is already provided. Just the CS's blown up on their own gives me very little information about what else the cell is doing in this condition.*

R3.26. We direct the reviewer to our response in R3.25.

C3.27. *Fig. S3, the merged images on the right of panels a and b are much too small. Also, please label the lobule(s) that is shown in panels c and d.*

R3.27. We have removed the DAPI panels for sake of space and clarity and have increased the sizes of the mCherry-ChR2 (magenta) and anti-calbindin (yellow) images in Supplementary figure 4 a.

Reviewer #2 (Remarks to the Author):

The manuscript by Sathyanesan and colleagues describes the long-term consequences of neonatal hypoxia for the young adult organism. For that purpose, they took advantage of an experimental mouse model in which, for a few days, perinatal mice are reared under hypoxic conditions (lowering oxygen concentration to 10 % rather than 20 %). This work is an extension of their previous study (Zonouzi et al., 2015).

Here, the authors focused on the analysis of behavioral alterations using automated assessment of motor behavior by the Erasmus ladder. The mice have to run along two rows of alternately positioned rungs. The automatic control senses not only sense the weight on individual rungs while the mouse is running along the ladder, but can also insert obstacles. Thereby, this device cannot only quantify impairment of motor performance, but also of associative and adaptive motor learning. These experiments were complemented by optogenetic and pharmacological experiments.

The main findings are:

- 1. Perinatal hypoxia induces a significant impairment of motor performance and of cerebellar learning.*
- 2. In young adult mice, the behavior can partially be rescued by (1) evoked neuronal photo-stimulation or (2) inhibition of GABA reuptake.*

While the first result might expected, the second result provides a strong translational aspect. Modulation of GABAergic signaling might be a potent target for future drug development and treatment of pre-term infants.

C4.1. The experiments are well performed, and the results are important. The manuscript itself is concisely written. The discussion section is very stimulating for future research.

R4.1. We thank the reviewer for their positive assessment of our manuscript.

C4.2. Although this work adds to a previous study, there are still open questions: Zonouzi et al. demonstrated impaired NG2 cell proliferation and differentiation, while here the physiology of Purkinje cells was studied. The authors left it to future studies to provide a solid experimental link between impaired NG2 and Purkinje cell activity.

R4.2. This is indeed an open and exciting question, and one which we are pursuing as part of a related project in the lab. Our scope in this manuscript involved the behavioral consequences of neonatal injury for which the most direct approach was to define physiological deficits at the level of the Purkinje cells themselves.

Minor points:

C4.3. Although the statistical values are given, a supplementary table listing the precise numbers of mice tested per individual experiment would be very helpful to estimate the statistical power. The authors do not provide scatter plots for all figures (e.g., they are missing in figure 2-4, and 7, but are depicted in figure 5 or for the optogenetic and pharmacological data sets).

R4.3. All our learning behavioral data are time-series data, some of which have four kinds of series (such as post-perturbation and pre-perturbation steptimes, in Nx and Hx). Hence, it would be confusing to the reader if we presented scatter plots along the time series. For the sake of clarity, we have left the original figures as is. However, for the purposes of peer-review, we have provided scatter plot data as Reviewer Figures 3-6.

Reviewer #3 (Remarks to the Author):

The authors have performed an in depth analysis of the behavioral consequences of cerebellar alterations resulting from hypoxic brain injury in the newborn, using a murine chronic hypoxia model.

They have used an Erasmus Ladder test to study cerebellar behavior and demonstrate that hypoxia leads to locomotor malperformance and also cerebellar learning deficits. They use an optogenetic approach for the first time in a chronic hypoxia model system to show that hypoxia leads to Purkinje cell firing frequency dysfunction. Moreover, they demonstrate that the GABA-reuptake inhibitor Tiagabine partially restores the defective locomotor performance and improves PC firing.

C5.1. This is a technically very good and thorough paper, introducing several new techniques, both behavioral and pathophysiological, for evaluating cerebellar dysfunction in the context of hypoxic injury.

R5.1. We thank the reviewer for their positive assessment of our manuscript.

C5.2. I would recommend a more detailed discussion of other non-cerebellar brain regions which might also be affected in hypoxia , and contribute to the locomotor dysfunction.

R5.2. We have added modified the text under discussion to include a paragraph on non-cerebellar brain regions which may contribute to locomotor dysfunction:

“A major caveat regarding our misstep data is the contribution of brain regions other than the cerebellum, such as the brainstem, limbic system, spinal cord, basal ganglia, and cerebral cortex⁶⁴ in modulating locomotor behavior. Since our animal model is a global

hypoxia model aiming to recapitulate most of the clinical hallmarks of neonatal birth injury, multiple brain regions associated with locomotor behavior could be potentially altered. Interestingly, while many studies identify the cerebellum as being particularly vulnerable during the perinatal brain injury time window, other locomotor-related brain regions such as the brainstem⁶⁵ and pre-motor regions⁶⁶ have also been suggested to undergo modest structural alterations associated with prematurity. Likely non-cerebellar candidate brain regions affected by neonatal brain injury could most likely comprise the major basal ganglia nuclei (caudate, putamen, nucleus accumbens, pallidum) and the mesencephalic locomotor region (MLR) of the brainstem, which have long been implicated in locomotor control. In fact, a recent longitudinal study indicates that smaller volumes of basal ganglia nuclei in very preterm infants are associated with long-term neurodevelopmental motor deficits at 7 years of age⁶⁷. This study also showed association of smaller neonatal thalamic volumes with motor deficits, suggesting that the entire cerebello-thalamo-cortical pathway⁶⁸ may be disrupted due to neonatal injury. Thus, although we cannot exclude the possibility that other brain regions might also be involved in the locomotor abnormalities observed in Hx mice, the cerebellum is an important component of the locomotor control circuit and plays a fundamental role in regulating this behavior.”

Reviewer #4 (Remarks to the Author):

C6.1. In this report, Sathyanesan and colleagues find that neonatal hypoxia exposure results in motor dysfunction and motor learning abnormalities that point to cerebellar injury. In vivo characterization of Purkinje cell (PC) activity revealed simple spiking deficits in hypoxic (Hx) mice. Behavioral and functional deficits could be partially reversed by pharmacological perturbation of the “developing GABA system”. The causal mechanisms underlying pharmacological

amelioration of PC spiking and motor behavior deficits remained unexplored. Provided the authors' previously reported observations of Hx-induced connectivity changes in the cerebellum and accompanying changes in PC intrinsic excitability (measured in vitro) (Zonouzi et al. 2016), the findings in this report are rather unsurprising. Perturbation of PC firing is well known to affect movement and motor learning. That multi-day treatment with a GABA uptake blocker (Tiagabine) can partially ameliorate the effects of neonatal Hx are, again, not surprising as the authors have already reported that such treatment reverses cellular-level effects of hypoxia on GABAergic circuitry in the cerebellum (Zonouzi et al. 2016). By missing an opportunity to provide further insight into how GABA uptake block leads to partial restoration of cerebellar function and its influence on behavior, the authors fail to provide useful information that may lead to the further development of therapeutic interventions for Hx-induced brain injury. Overall, this work is unlikely to find a wide audience outside of the hypoxia field. Also, the work suffers from several potential issues with experimental design.

R6.1. We thank the reviewer for their candid and critical assessment of our manuscript. The scope of this manuscript comprised the identification of behavioral and *in vivo* physiological abnormalities in a translational and clinically relevant animal model of neonatal brain injury. We acknowledge that we have not definitively addressed “causal mechanisms”, although we think that our pharmacological intervention with Tiagabine provides a promising first-pass at mechanistic details, which is a subject of ongoing investigation in our lab.

Regarding our work not finding a wide audience outside the hypoxia field: we respectfully disagree. We have presented our work at the Gordon cerebellum conference 2017 and have received positive assessments of our work. Additionally, Dr. Aaron Sathyanesan, the first author of this manuscript was selected to chair one of only two nanosymposia on the cerebellum entitled “Cerebellum: from circuitry to function” at the Society for Neuroscience (SfN) 2017 Annual

Meeting held in Washington DC. Dr. Sathyanesan's abstract (which was based on this manuscript) was also selected for an oral presentation at this nanosymposium.

C6.2. Motor deficits are presumably due to PC spiking changes DURING motor behavior. However, the effects of Hx are only explored in vivo on intrinsic excitability under anesthesia (how is this measure really different than that that already performed in slices by the same group?). This is a major limit to the conclusions drawn in the present work especially if motor performance and learning deficits are to be related to PC spiking abnormalities.

R6.2. As we mentioned earlier, the current design of the commercially available Erasmus Ladder restricts us from performing recordings as the animal is behaving on the ladder. However, we are working with Noldus Inc. on designing a prototype for performing exactly the kind of experiments which the reviewer is suggesting. However, this is a potential future experiment. Having said this, however, we believe that while *in vivo* multielectrode recordings from anesthetized animals is not as powerful as realtime recordings from behaving animals, it is certainly not on the same plane of relevance as *in vitro* electrophysiological analysis. Although each technique has its own strengths and weaknesses. Regarding similarities to our earlier work¹⁰ - while there may be some tangential similarities e.g. PC firing frequency, this was not the main objective of our previous work, which dealt more with GABAergic interneuron-NG2 cell interactions in the cerebellar white matter.

C6.3. Regarding measurements of ChR2-induced spiking in PCs, the authors face a potential confound in that they have not adequately controlled for Hx-induced changes in ChR2 expression (perhaps neonatal hypoxia exposure diminishes ChR2 expression levels in PCs compared to Nx controls). Although the authors attempt to control for this possibility by looking at mCherry

fluorescence, this is an indirect method. A more well-controlled experiment would be to measure evoked photocurrent amplitudes in Hx and Nx treated mice.

R6.3. In the revised manuscript, we have provided *in vitro* slice electrophysiology data which show that Hx PCs are viable (Supplementary figure 2a-f) and respond to optogenetic stimulation (Supplementary figure 2g-j)

C6.4. *Complex spikes are evoked by the activity of climbing fibers. So why does optogenetic stimulation of PCs induce complex spiking? An indirect effect through disinhibition of the inferior olive would be expected to be accompanied by a delay- this is not apparent (or quantified) in their data.*

R6.4. We have provided representative traces of complex spikes in Figure 9 a-b (complex spikes marked by asterisks). It is quite likely that the increase we see under optostimulation is due to disinhibition of the inferior olive. Figure 9 d-e have been plotted using 100 ms bin width, which is within the range of the delay suggested by Chaumont et al (2013)¹⁸.

C6.5. *If diminished PC simple spike output is a key variable explaining Hx-induced locomotion and motor learning deficits, can this be recovered by optogenetic stimulation of PCs during behavior?*

R6.5. We would like to direct the reviewer to R6.2.

C6.6. *Is the Hx effect on PC excitability ubiquitous across the cerebellum? PCs in zebrin-delimited zones show extensive functional differences including in their intrinsic excitability. This*

should be assessed. Relatedly, are the cerebellar areas that were recorded from even related to the motor behaviors studied in their task?

R6.6. All our electrophysiological recordings were consistently performed on the simple lobule (Supplementary figure 4 b-c insets) in every single experiment. We have not explored the effect of Hx on other lobules. In order to assess the effect of Zebrin-delimited zones on firing frequency, we performed post-hoc immunohistochemistry after completion of *in vivo* electrophysiological recordings. We stained for anti-PLC β 4 which is a “bridge-marker” labeling Zebrin-negative zones (Reviewer figure 7, yellow). It was challenging to definitively conclude if the electrode position (Reviewer figure 7, white triangle) landed in a Zebrin-positive (PLC β 4-positive) or Zebrin-negative (PLC β 4-positive) due to autofluorescence (black asterisks). However, the simple lobule itself has both Zebrin-positive and Zebrin-negative PCs (white boxes, magnified view presented in the side panels: anti-calbindin – magenta, anti-PLC β 4 – yellow, merge). Since we sample different aspects of the simple lobule, it is quite likely that we are sampling both Zebrin positive and negative populations of PCs. Although there is no prior data regarding the function of the simple lobule in the Erasmus Ladder task, the simple lobule has been implicated in associative forelimb movement learning¹⁴.

C6.7. *The Pcp2-Cre line used in the study should be identified (Jackson Lab maintains two different lines; Jdhu and Mpin). Notably, Cre activity in the Mpin line has been shown to be non-specific for PCs (see Witter et al. and Regehr 2016 [Neuron] who used functional assessments with ChR2-induced excitation which is more sensitive than an XFP reporter as used in the present study; also see Zhang et al. 2004 [Genesis]). If the authors used the Mpin line, the lack of PC specificity could influence the outcome of their results.*

R6.7. In the revised manuscript, we have added immunohistochemical and cell density comparisons in Nx and Hx groups showing exceedingly rare instances of non-specific labeling of putative non-PCs (Supplementary figure 3). Considering the effect sizes we observe, it is unlikely that these rare events significantly affect the interpretation of our results. We have mentioned the results from this analysis under the methods section:

Finally, to identify the extent of non-specific viral labeling of cells in the cerebellar cortex, we counted mCherry+/calbindin- cells in the molecular layer. Confirming an earlier report²⁵, we did notice non-specific viral expression in non-PC cells in the molecular layer in both Nx and Hx groups (Supplementary figure 3, a-b, arrowhead). However, using our particular injection protocol which targeted the the simple lobule, this non-specific labeling was exceedingly rare (2 non-PCs:133 PCs from n = 3 animals in Nx and 4 non-PCs:161 PCs from n = 3 animals; < 0.1 cells/ $[(100 \mu\text{m})^3]$ in Nx and < 0.2 cells/ $[(100 \mu\text{m})^3]$ in Hx;) and not significantly different between Hx and Nx groups (Supplementary figure 3c, Unpaired t-test, two-tailed, $t = 0.7686$, $df = 4$, $P = 0.4850$; Supplementary figure 3d, Fisher's exact test, two-tailed, $P = 0.6938$).

C6.8. Notably, the authors observe recovery over time for locomotor deficits (i.e. at P45). Are compensatory mechanisms for this recovery observable at the circuit level? If so, do they match that induced by Tiagabine (e.g., are PC simple spiking deficits restored)?

R6.8. In our revised manuscript, we have added a fresh set of experiments for all P45 groups in order to maintain consistency (Supplementary figure 6). The newly added electrophysiological data shows that there is partial recovery of Purkinje cells firing activity over time even at P45. Tiagabine facilitates Purkinje cell simple spikes firing activity to some extent (but not completely). This suggests that there may be a long-term defect at the PC circuit-level that manifests as a

delayed learning problem, and that Tiagabine may not be directly affecting the PC-learning circuitry.

C6.9. Is the effect of Tiagabine on ameliorating motor deficits specific for the cerebellum (the authors use IP injection of Tiagabine which will have a global influence)? Why not apply Tiagabine locally (e.g., by cannulae) to bolster the conclusion that recovery is specific for this brain area?

R6.9. This is a possibility. Applying Tiagabine locally by cannulae would be helpful identifying cerebellum-specificity in terms of pharmacological intervention, however, we preferred using Tiagabine through a clinically relevant route of delivery for human neonates and infants. Our objective behind using Tiagabine was that it is routinely used in pediatric clinical treatment.

C6.10. How is Tiagabine working to partially restore PC spiking? How does this relate to “targeting the developmental GABA system”? Is there an increase or decrease in GABA levels with this pharmacological manipulation? Is GABA uptake block affecting cortical GABAergic circuitry (OPCs, Golgi cells, stellate/basket cells)? Purkinje cells themselves are also GABAergic so GABA re-uptake block in the deep cerebellar nuclei might also effect their influence on behavior.

R6.10. It is quite possible that Tiagabine works on multiple GABAergic celltypes. We direct the reviewer to the discussion section of our manuscript. Future experiments would need to address the mechanistic nitty-gritty of Tiagabine’s precise action on subtypes of cells in the cerebellum.

C6.11. Presumptively, experience-dependent improvement in misstep performance for both Nx and Hx mice during the initial training phase (session 1-4) reflects some type brain-dependent function. Is the locus known?

R6.11. No, the locus for this initial improvement in misstep performance is not known.

References

1. van Welie I, Smith IT, Watt AJ. The metamorphosis of the developing cerebellar microcircuit. *Current opinion in neurobiology* **21**, 245-253 (2011).
2. Eilers J, Plant TD, Marandi N, Konnerth A. GABA-mediated Ca²⁺ signalling in developing rat cerebellar Purkinje neurones. *The Journal of physiology* **536**, 429-437 (2001).
3. Watt AJ, Cuntz H, Mori M, Nusser Z, Sjöström PJ, Häusser M. Traveling waves in developing cerebellar cortex mediated by asymmetrical Purkinje cell connectivity. *Nature neuroscience* **12**, 463-473 (2009).
4. Chavas J, Marty A. Coexistence of excitatory and inhibitory GABA synapses in the cerebellar interneuron network. *The Journal of neuroscience : the official journal of the Society for Neuroscience* **23**, 2019-2031 (2003).
5. Brickley SG, Cull-Candy SG, Farrant M. Development of a tonic form of synaptic inhibition in rat cerebellar granule cells resulting from persistent activation of GABA_A receptors. *The Journal of physiology* **497 (Pt 3)**, 753-759 (1996).
6. Barbato C, *et al.* MicroRNA-92 modulates K(+) Cl(-) co-transporter KCC2 expression in cerebellar granule neurons. *Journal of neurochemistry* **113**, 591-600 (2010).
7. Heiney SA, Wohl MP, Chettih SN, Ruffolo LI, Medina JF. Cerebellar-dependent expression of motor learning during eyeblink conditioning in head-fixed mice. *The Journal of neuroscience : the official journal of the Society for Neuroscience* **34**, 14845-14853 (2014).
8. Medina JF, Lisberger SG. Links from complex spikes to local plasticity and motor learning in the cerebellum of awake-behaving monkeys. *Nature neuroscience* **11**, 1185-1192 (2008).
9. Kloth AD, *et al.* Cerebellar associative sensory learning defects in five mouse autism models. *eLife* **4**, e06085 (2015).
10. Zonouzi M, *et al.* GABAergic regulation of cerebellar NG2 cell development is altered in perinatal white matter injury. *Nature neuroscience* **18**, 674-682 (2015).
11. Tye KM, Deisseroth K. Optogenetic investigation of neural circuits underlying brain disease in animal models. *Nature reviews Neuroscience* **13**, 251-266 (2012).

12. Beck H, Yaari Y. Plasticity of intrinsic neuronal properties in CNS disorders. *Nature reviews Neuroscience* **9**, 357-369 (2008).
13. Chen S, Mohajerani MH, Xie Y, Murphy TH. Optogenetic analysis of neuronal excitability during global ischemia reveals selective deficits in sensory processing following reperfusion in mouse cortex. *The Journal of neuroscience : the official journal of the Society for Neuroscience* **32**, 13510-13519 (2012).
14. Lee KH, *et al.* Circuit mechanisms underlying motor memory formation in the cerebellum. *Neuron* **86**, 529-540 (2015).
15. Chihabi K, Morielli AD, Green JT. Intracerebellar infusion of the protein kinase M zeta (PKMzeta) inhibitor zeta-inhibitory peptide (ZIP) disrupts eyeblink classical conditioning. *Behavioral neuroscience* **130**, 563-571 (2016).
16. Nicholson DA, Freeman JH, Jr. Developmental changes in eyeblink conditioning and simple spike activity in the cerebellar cortex. *Developmental psychobiology* **44**, 45-57 (2004).
17. Zhang H, Cohen AE. Optogenetic Approaches to Drug Discovery in Neuroscience and Beyond. *Trends in biotechnology* **35**, 625-639 (2017).
18. Chaumont J, *et al.* Clusters of cerebellar Purkinje cells control their afferent climbing fiber discharge. *Proceedings of the National Academy of Sciences of the United States of America* **110**, 16223-16228 (2013).
19. Valeeva G, Tressard T, Mukhtarov M, Baude A, Khazipov R. An Optogenetic Approach for Investigation of Excitatory and Inhibitory Network GABA Actions in Mice Expressing Channelrhodopsin-2 in GABAergic Neurons. *The Journal of neuroscience : the official journal of the Society for Neuroscience* **36**, 5961-5973 (2016).
20. Bitzenhofer SH, *et al.* Layer-specific optogenetic activation of pyramidal neurons causes beta-gamma entrainment of neonatal networks. *Nature communications* **8**, 14563 (2017).
21. Rudy B, McBain CJ. Kv3 channels: voltage-gated K⁺ channels designed for high-frequency repetitive firing. *Trends in neurosciences* **24**, 517-526 (2001).
22. Hurlock EC, Bose M, Pierce G, Joho RH. Rescue of motor coordination by Purkinje cell-targeted restoration of Kv3.3 channels in Kcnc3-null mice requires Kcnc1. *The Journal of neuroscience : the official journal of the Society for Neuroscience* **29**, 15735-15744 (2009).

23. Hurlock EC, McMahon A, Joho RH. Purkinje-cell-restricted restoration of Kv3.3 function restores complex spikes and rescues motor coordination in *Kcnc3* mutants. *The Journal of neuroscience : the official journal of the Society for Neuroscience* **28**, 4640-4648 (2008).
24. Hoogland TM, De Gruijl JR, Witter L, Canto CB, De Zeeuw CI. Role of Synchronous Activation of Cerebellar Purkinje Cell Ensembles in Multi-joint Movement Control. *Current biology : CB* **25**, 1157-1165 (2015).
25. Witter L, Rudolph S, Pressler RT, Lahlaf SI, Regehr WG. Purkinje Cell Collaterals Enable Output Signals from the Cerebellar Cortex to Feed Back to Purkinje Cells and Interneurons. *Neuron* **91**, 312-319 (2016).

Reviewer figure 1. (a) Table displays the number of channels/electrodes used for recording along with the total number of waveforms recorded in each electrode. Using a specific threshold cutoff, the number of sorted/selected spikes were marked as 'unit a' and 'unit b' from respective electrode. (b) (e) and (h) show raw traces recorded by the respective electrodes (color matched) along with the principal component analysis in (c), (f) and (i) showing separation of sorted spike clouds (colored) from noise (grey) in 2D plane and the corresponding sorted waveforms in (d) (g) and (j) (colored) overlaid on unsorted components (grey) Scale 100 μ s. Using the selection criteria (mentioned in methods), the blue waveforms in panel (g) were identified as Purkinje cell simple spikes from P13 Nx mice.

Reviewer figure 2. (a) A representative raw trace recorded from P21 Nx mouse; complex spikes are denoted by asterisks (b) Simple spikes and (e) complex spikes were consistently isolated in Offline Sorter (Plexon V4) by applying different cutoff threshold values as shown as red line in continuous waveforms. Combined with other selection criteria for both simple spikes and complex spikes (detailed in methods) sorted spikes are identified: magenta – simple spike, blue – complex spike. PCA separation of (c) sorted simple spikes and (f) sorted complex spikes is shown, with grey (both panels) being the unsorted waveforms. Scale in panels ‘d’ 100 μ s, ‘g’ – 200 μ s.

a

b

Reviewer figure 3. (a) Misstep data from figure 2 presented as scatter plots per session (open circles – individual measurement, black – Nx mice, red – Hx mice, solid circle represent mean value (b) backstep data from figure 2

Reviewer figure 4. (a) Post-perturbation step time data from figure 3a presented as scatter plots per session (open circles – individual measurement, black – Nx mice, red – Hx mice, solid circle represent mean value) (b) Normalized learning data from 3b (c) extended learning data from figure 3c shown as before-after plot per session.

a

b

Reviewer figure 5. (a) Post-perturbation step time data from figure 4a presented as scatter plots per session (open circles – individual measurement, black – Nx mice, red – Hx mice, solid circle represent mean value (b) Normalized learning data from 4b

a

b

Reviewer figure 6. (a) Misstep data from figure 7a presented as scatter plots per session (open circles – individual measurement, green – Hx + Tiagabine mice, red – Hx + saline mice, solid circles represent mean value (b) backstep data from 3b plotted as scatter plot

Reviewer figure 7. (a) Representative confocal images of simple lobule following post-hoc immunohistochemistry following *in vivo* multielectrode recording from Nx mouse and (b) Hx mouse (anti-calbindin – magenta, anti-PLCβ4 – yellow, DAPI – blue, white arrowhead represents electrode position, black asterisks mark regions of excessive autofluorescence, likely because PLCβ4 antibody is a mouse monoclonal antibody) panels to the left of larger composite represent areas highlighted by white box. Note the presence of both PLCβ4⁺ (putatively Zebrin⁻) and PLCβ4⁻ (putatively Zebrin⁺) PCs in the simple lobule from which we typically survey different cortical regions via *in vivo* electrophysiology. Scale in larger composite – 100 μm, scale in magnified panels – 25 μm.

Reviewer figure 8. Representative current response from PC from Nx + ChR2 animal (black) and Hx + ChR2 animal (red) under voltage clamp mode and continuous optostimulation in *in vitro* slice conditions.

Reviewers' comments:

Reviewer #1 (Remarks to the Author):

The authors have submitted a revised version of their manuscript on neonatal hypoxia in cerebellum. The authors have provided an in depth response to every single comment, and addressed the comments by revising the text and in some cases providing additional data. The authors have done an excellent job in addressing the Reviewer comments and I appreciate the time and effort that was taken to explain each response carefully.

This is an exciting manuscript with important information for all neuroscientists, and even those with interests beyond brain science.

I do not have any further comments.

Reviewer #2 (Remarks to the Author):

The authors submitted a revised version of their manuscript. The revision and the accompanying rebuttal letter address all points raised by this reviewer.

Reviewer #4 (Remarks to the Author):

In this revised manuscript, Sathyanesan and colleagues continue to report that early exposure to hypoxia generates damage to the cerebellum that results in motor deficiencies that can, in part, be ameliorated by GABA-reuptake inhibitors. Although the authors included new data to address the concerns of other reviewers, they were largely dismissive of my own points. Though they acknowledge that their experiments have many shortcomings/problems as I pointed out, they indicate that they don't have the technical expertise to address these issues. Acknowledging a problem but doing nothing about it doesn't solve it. In addition, the few points that they attempted to address were performed in a marginal way at best. For example, they now confirm that the Purkinje cell Cre mouse line that they used for their experiments is not specific for Purkinje cells. They include histochemical data showing low expression levels in cells in the molecular layer (notably, this new data is reported in the Methods section). As I stated in my initial assessment, histochemical analysis will under-represent the extent of off target expression therefore a functional assessment is required (using electrophysiological recordings to characterize this same mouse line used by Sathyanesan, the Regehr group found optogenetic-induced responses in >20% of these molecular layer cells). In conclusion, the authors did little to address my concerns; stronger consideration of these points is warranted.

Neonatal brain injury causes cerebellar learning deficits and Purkinje cell dysfunction
By A. Sathyanesan, S. Kundu, J. Abbah, and V. Gallo

Response to Reviewers Comments

We thank the reviewers for their earlier comments and criticism (round 1 of revisions), which has served to significantly strengthen our manuscript entitled “Neonatal brain injury causes cerebellar learning deficits and Purkinje cell dysfunction” by Sathyanesan, Kundu, Abbah, and Gallo. We would like to acknowledge the acceptance of our revisions by Reviewers #2, and #3. In this further-revised manuscript, we have addressed the suggestion and criticism by Reviewer #4. Specifically, we have addressed: 1) the issue of cell type specificity in the *Pcp2-Cre* line which we have used (Mpin line), and 2) the possibility of evoked complex spikes in our *in vivo* electrophysiology data being an experimental artifact. As a result, we have included one additional supplemental figure and one reviewer figure to address these criticisms. Below, we have included a detailed response to particular criticisms by Reviewer #4.

1. Non-specific targeting of molecular layer interneurons (MLIs) in the Pcp2-Cre Mpin line (Witter et al., 2016)

In our first round of revisions, we had shown immunohistochemistry and cell-count data that there is an exceedingly low percentage of off-target recombination in non-PCs which presumptively may be MLIs (Supplementary Figure 3 in the current revised manuscript). Witter et al (2016) crossed a conditional channelrhodopsin mouse line (ChR2-EYFP (B6;129S-Gt(ROSA)26Sor^{tm32(CAG-COP4*H134R/EYFP)Hze}) with the *Pcp2-Cre* Jdhu and Mpin lines, and present data supporting the claim that the Jdhu line is more specific to PCs than the Mpin line. Specifically, per Witter et al (2016), there is a direct light-evoked photocurrent in MLIs in the Mpin line. In Figure 3, panel e of their paper, Witter et al show a representative light-evoked photocurrent in an MLI, with very short latency (< approx. 5 ms). Indeed, this is a robust off-target response from MLIs in their data.

In all our electrophysiology experiments, we have used viral injections of ChR2 (AAV9.EF1.dflox.hChR2[H134R]-mCherry.WPRE.hGH) in a specific location of the cerebellar cortex of *Pcp2-Cre* Mpin line, which results in mCherry⁺ AAV-labeled cells. We attempted to record from the rare instance of mCherry⁺ non-PC using optical stimulation in slice electrophysiology experiments. However, despite repeated and focused efforts, we were unable to locate any mCherry⁺ non-PC to record from. This further validates our data in Supplementary Figure 3 that mCherry⁺ non-PCs are very rare.

In their article, Witter et al (2016) further claim:
“21% of the MLIs (6 of 29) had photocurrents in the Mpin line” (Results section, p 315, Witter et al., 2016)

Assuming that the percentage of viral-vector carrying non-PCs we observed in fixed post-electrophysiology immunohistochemistry data is equivalent to the 3% of tdTomato⁺ MLIs shown by Witter et al (2016) (obtained by crossing the Mpin line to a tdTomato reporter line), as the reviewer points out, there is still a possibility that mCherry⁻ MLIs in the vicinity of mCherry⁺ PCs may show direct photocurrents. If we make the assumption that the Witter et al (2016) experimental design is equivalent to our design, then this would mean that, even in our experiments, ~21% of MLIs (irrespective of mCherry expression) should respond with a direct and robust photocurrent when stimulated. To address this possibility, we performed recordings from randomly picked MLIs in the

vicinity of the mCherry⁺ PCs in conjunction with optical stimulation. We found that optostimulation did not elicit a photocurrent in all MLIs tested (0% i.e. 0/19 cells from 5 animals, individual traces for all MLIs shown in Supplementary figure 4 b). In their paper, Witter et al (2016) found that the mean photocurrent in MLIs in Pcp2-Cre-Mpin-ChR2-EYFP animals is ~150 pA (Figure 3 g, Witter et al., 2016). In our recordings from MLIs, we did not observe any direct and robust response to optostimulation (Supplementary figure 4 b). To ensure that MLIs tested were healthy and responsive in general, we used a standard current injection protocol (Supplementary figure 4 c) after the optostimulation.

Thus, our data strongly suggests that MLIs in the vicinity of mCherry⁺ PCs in our AAV-injected Mpin animals do not show a direct and robust photocurrent in response to optical stimulation. It is quite possible that the differences in non-specific responses between our data and Witter et al (2016) have to do with the differences in experimental design, particularly regarding our use of viral injections, compared to the use of a conditional ChR2 transgenic mouse by Witter et al (2016). While both techniques have their specific pitfalls and advantages, our use of viral injection into the Pcp2-Cre Mpin line may have circumvented the issue of increased “leaky” Cre recombinase activity in non-PCs.

2. Evoked complex spikes (CSs) may be a result of artefactual optical stimulation of PC dendrites

In order to address the possibility that evoked CSs in our *in vivo* electrophysiology data are an artifact of dendritic stimulation, we have conducted a latency analysis of our spike data (Reviewer Figure 1). If the evoked CSs are an artifact of dendritic stimulation, one would predict that the latency to ‘spike’ artifact from light onset would be very low (< 5 ms) (Laxpati et al., 2014). However, our latency analysis shows that the median latency for evoked CSs is 139 ms (Reviewer Figure 1b). Comparatively, median evoked simple spike latency is lower – 6 ms (Reviewer Figure 1a). The median latencies of SSs and CSs which we observe is on the order of latency to first spike in previously published reports (Tsubota et al., 2011; Chaumont et al., 2013; Kruse et al., 2014; Witter et al., 2013). It is helpful to emphasize that our study is not the first to observe an increase in CS frequency during optical stimulation *in vivo* in PCs expressing ChR2. The median latency from light onset to increase in CS frequency that we observe is comparable to previous studies (Chaumont et al., 2013; Witter et al., 2013).

3. Specificity of Tiagabine using cannulation experiments

The age at which we are targeting pharmacological intervention in C57BL/6 Hx animals (P12-P16) makes it technically unfeasible to incorporate cannulation to deliver Tiagabine directly to the cerebellum. Other than this technical issue, the reasons we have not included cannulation experiments in our study is as follows:

1. We are currently examining a number of more targeted approaches, including those which are pharmacology- and genetics-based.
2. However, as is, the Tiagabine data is important because this shows that pharmacological treatment with a drug that is currently used in the clinic is a viable option to treat cerebellar-mediated behavioral deficits due to neonatal brain injury.
3. This is a set of experiments that links this study to our previous paper in *Nature Neuroscience* (Zonouzi et al., 2015)

In summary, we think that the proposed cannulation experiment is beyond the scope of this study.

References

1. Witter L, Rudolph S, Pressler RT, Lahlaf SI, Regehr WG. Purkinje Cell Collaterals Enable Output Signals from the Cerebellar Cortex to Feed Back to Purkinje Cells and Interneurons. *Neuron* **91**, 312-319 (2016).
2. Laxpati NG, Mahmoudi B, Gutekunst CA, Newman JP, Zeller-Townson R, Gross RE. Real-time in vivo optogenetic neuromodulation and multielectrode electrophysiologic recording with NeuroRighter. *Frontiers in neuroengineering* **7**, 40 (2014).
3. Tsubota T, Ohashi Y, Tamura K, Sato A, Miyashita Y. Optogenetic manipulation of cerebellar Purkinje cell activity in vivo. *PloS one* **6**, e22400 (2011).
4. Chaumont J, *et al.* Clusters of cerebellar Purkinje cells control their afferent climbing fiber discharge. *Proceedings of the National Academy of Sciences of the United States of America* **110**, 16223-16228 (2013).
5. Kruse W, Krause M, Aarse J, Mark MD, Manahan-Vaughan D, Herlitze S. Optogenetic modulation and multi-electrode analysis of cerebellar networks in vivo. *PloS one* **9**, e105589 (2014).
6. Witter L, Canto CB, Hoogland TM, de Gruijl JR, De Zeeuw CI. Strength and timing of motor responses mediated by rebound firing in the cerebellar nuclei after Purkinje cell activation. *Frontiers in neural circuits* **7**, 133 (2013).
7. Zonouzi M, *et al.* GABAergic regulation of cerebellar NG2 cell development is altered in perinatal white matter injury. *Nature neuroscience* **18**, 674-682 (2015).

Reviewer Figure 1. Evoked CSs have increased latency compared to evoked SSs. a. Population mean spike counts for PC SSs at P30 (grey columns). 'X' axis has been plotted with a temporal resolution of 1 ms for the window -200 ms to +200 ms where $t = 0$ is onset of optostimulation (blue bars) b. Population mean spike counts for PC CSs at P30 (black columns). c. SS (grey) and CS (black) scatter plot comparison of latency to first spike after optostimulation onset. Long black horizontal line denotes mean latency; whiskers denote \pm SEM ($n = 29$ PCs for 8-10 animals for both groups in all panels); blue line denotes median latency: SS = 6 ms, CS = 139 ms. Asterisk represents $P < 0.0001$ (exact) for Two-tailed Mann-Whitney U test. Difference between medians – Actual = 133, Hodges-Lehmann = 133.

REVIEWERS' COMMENTS:

Reviewer #4 (Remarks to the Author):

The authors have adequately addressed my concerns.